# Assessing Mechanical Properties and Response of Expansive Soft Rock in Tunnel Excavation: A Numerical Simulation Study

**DOI:** 10.3390/ma17081747

**Published:** 2024-04-11

**Authors:** Hao Ma, Youliang Chen, Lixin Chang, Xi Du, Tomas Manuel Fernandez-Steeger, Dongpeng Wu, Rafig Azzam, Yi Li

**Affiliations:** 1Department of Civil Engineering, School of Environment and Architecture, University of Shanghai for Science and Technology, Shanghai 200093, China; mahao9881@163.com (H.M.); duxijl@163.com (X.D.); dongpeng_w@163.com (D.W.); liyi950526@163.com (Y.L.); 2Institute of Architectural Engineering, Shanghai Zhongqiao Vocational and Technical University, Shanghai 201514, China; changlixiniukl@163.com; 3Institut für Angewandte Geowissenschaften, Technische Universität Berlin, 10587 Berlin, Germany; fernandez-steeger@tu-berlin.de; 4Department of Engineering Geology and Hydrogeology, RWTH Aachen University, 52056 Aachen, Germany; azzam@lih.rwth-aachen.de

**Keywords:** numerical investigation, tunnel, humidity diffusion, strength criteria

## Abstract

This study investigates the dynamics of moisture absorption and swelling in soft rock during tunnel excavation, emphasizing the response to support resistance. Utilizing COMSOL numerical simulations, we conduct a comparative analysis of various strength criteria and non-associated flow rules. The results demonstrate that the Mohr–Coulomb criterion combined with the Drucker–Prager model under compressive loads imposes stricter limitations on water absorption and expansion than when paired with the Drucker–Prager model under tensile loads. Restricted rock expansion leads to decreased horizontal displacement and ground uplift, increased displacement in the tunnel’s bottom arch, and significantly reduced displacement in the top arch. The study also considers the effects of shear dilation, burial depth, and support resistance on the stress and displacement of the surrounding rock. Increased shear dilation angles correlate with greater rock expansion, resulting in increased horizontal displacement and ground uplift. The research study concludes that support resistance is critical in limiting the movement of the tunnel’s bottom arch and impacting the stability of the surrounding rock. Additionally, the extent of rock damage during the excavation of expansive soft rock tunnels is found to be minimal. Overall, this study provides valuable insights into the processes of soft rock tunnel excavation and contributes to the development of more efficient support systems.

## 1. Introduction

The intricate interaction between water and expansive soft rocks, as investigated in various studies, leads to significant volumetric expansion and changes in the mechanical properties of these materials, often resulting in severe damage to a range of structures such as water engineering facilities, underground chambers, and buildings [1,2,3,4,5,6,7]. This phenomenon is particularly pronounced when tunneling activities intersect with deeply buried, weak rock layers rich in expansive clay minerals. The process of excavation in these scenarios triggers a rapid release of stored energy within the original rock matrix, yielding a dual effect: a redistribution of stress within the surrounding rock mass that leads to alterations in its state and the induction of a swelling effect [8,9,10,11] and the emergence of unloading surfaces that facilitate the development and extension of expansion strain.

The progression of these phenomena is further complicated as water continuously infiltrates and diffuses into the rock mass during excavation, especially in areas with high potential for expansion. This infiltration causes significant inward displacement and uplift of tunnel walls and floors. In the absence of timely and adequate support measures, this can lead to the extensive deformation and failure of both the tunnel lining and the surrounding rock mass. The extent and degree of the moisture-affected rock mass are variable; the expansion and softening of the rock mass upon water interaction can induce comprehensive changes in stress and strain throughout the entire rock mass. Additionally, the stress field within the surrounding rock mass interacts with the moisture-affected portion, further complicating the overall geomechanical response.

To address these challenges, Miao et al. [12,13] proposed a conceptual framework for the humidity stress field theory, which accounts for the changes in stress and strain due to internal and external constraints in expansive surrounding rock masses. This theoretical framework has been substantiated and applied in various research contexts, encompassing theoretical foundations, empirical studies, and simulation-based explorations. Bai et al. [14] provided a rigorous validation of the humidity stress field theory, delving into its mechanical implications and conditions of applicability. Building upon this theoretical foundation, Zhu et al. [15] considered the variations in moisture content and their impact on the mechanical properties of expansive rocks, such as the elastic modulus, Poisson’s ratio, and yield limit, proposing an elastoplastic constitutive model tailored for these materials. Further empirical support for this theory was provided through free swelling tests on argillaceous shale conducted by Ji et al. [16,17,18], corroborating the theory’s predictions. Additionally, Wang et al. [19] developed an advanced elastoplastic constitutive model for the humidity stress field of expansive rocks based on incremental theory and expansion deformation mechanisms and integrated this model into the numerical software for practical applications in tunneling engineering.

Numerical simulations on tunnel excavation have been studied by many experts. Huang et al. [20] studied the influence of tunnel relative position on excavation, tunnel diameter, and excavation size through numerical simulations. Ng et al. [21] investigated the effect of tunnel position on tunnel deformation during the excavation process based on three-dimensional numerical analysis. Zheng et al. [22] conducted a large number of finite element simulations and obtained the characteristics of tunnel deformation caused by excavation. The research results indicate that the deformation of existing tunnels is significantly influenced by the deformation mode of the support structure. Shi et al. [23] studied the geometric impact of basement excavation on adjacent tunnels through a series of numerical models and found that compared to other shapes of basements, a short rectangular basement is a better excavation geometric shape. Zheng et al. [24] performed numerical simulations to study the deformation of tunnels intersecting at an oblique angle with the excavation. The results showed that the maximum lateral displacement occurred at the midpoint of the tunnel closest to the excavation boundary.

Tan [25] identified the primary causes of deformation and failure in tunnels excavated through expansive rocks: the synergistic effects of rock swelling due to water absorption and the expansion under deviatoric stress. These interrelated factors evolve over time, continuously affecting tunnel stability. Building on this understanding, our previous research derived a formula for moisture distribution in deeply buried expansive rock masses and ascertained the post-excavation stress distribution of surrounding rock in tunnels under the influence of the humidity field. This body of work contributes significantly to the understanding of the complex interactions between rock mechanics and hydrogeology in tunnel engineering, providing a foundation for the development of more effective design and construction strategies in challenging geotechnical environments.

This research focuses on tunnels embedded in deeply buried, expansive rock formations. Our primary concern is the evolution of rock fractures, a key aspect in geotechnical engineering. This is achieved through the application of numerical simulation methods. These methods are categorized into continuum and discrete approaches. We primarily utilize continuum methods, notably the finite element method (FEM), extended finite element method (XFEM), and finite difference method (FDM), as detailed by Liu et al. [26,27,28]. In summary, there have been many studies on the theoretical stress field of expansive surrounding rock. However, in terms of strength criteria, most studies use the Mohr–Coulomb criterion and its related flow rules. Therefore, employing COMSOL Multiphysics 6.1 software due to its ease of use, technical support, regular updates, and widespread usage, our study investigates the effects of expansion stress, shear dilation, burial depth, and support resistance on both the displacement and displacement fields. This exploration is conducted within the ambit of the Mohr–Coulomb strength criterion, accompanied by an evaluation of diverse flow rules to better understand their impact on these geological structures. The primary aims of this comprehensive study are to enhance the understanding of tunnel stability and integrity in complex rock formations and to provide theoretical support for actual tunnel excavation projects and stability analysis in order to better understand the data.

## 2. Average Ground Stress Model

### 2.1. Model Size

In the existing case model, the plastic zone formed by tunnel excavation is concentrated around the tunnel, so we set a relatively small region for better observation. For this study, we have constructed a two-dimensional graphical representation encompassing an area of 45 m by 45 m. Central to this graphical model is a semicircular feature, possessing a radius of 6 m, strategically positioned at the midpoint of the left boundary and maintained at a distance of 20 m from the upper boundary. After its construction, the graph undergoes a gridding process for enhanced analytical precision. Similarly, to achieve precise calculations and better observation, we align the physical field to set up a highly refined mesh. Furthermore, a two-dimensional coordinate system is meticulously established within this model. The upper boundary of the model is meticulously aligned with the *x*-axis, which is designated as the ground-level reference. Concurrently, the left boundary, demarcating the tunnel’s inception, is aligned along the *y*-axis. Orientation-wise, the positive *x*-axis extends towards the right while the positive *y*-axis is directed upwards, ensuring a standard Cartesian coordinate system. Leveraging the capabilities of the COMSOL Multiphysics software, a symmetrical boundary condition is implemented along the left boundary of the model. This computational approach facilitates the representation of the circular tunnel, which is conceptualized as being buried at a depth of 20 m. Such a simulation setup is crucial for analyzing the geomechanical interactions and stress distributions around the tunnel structure, providing valuable insights into the underground behavior in response to various geotechnical factors.

### 2.2. Model Parameters and Calculation Results

For this model, the model parameters are shown in the following Table 1.

The parameters include Young’s modulus (*E*), Poisson’s ratio (*v*), density (ρ), cohesion (*c*), the internal friction angle (*φ*), the humidity expansion coefficient (*α*), the maximum water content ($\omega_{max}$), and the initial water content ($\omega_{0}$).

Regarding boundary conditions, detailed discussions have been conducted by scholars [29,30]. In the two-dimensional model in COMSOL, the upper boundary is a free boundary representing the ground, the left boundary is a symmetric boundary used to reduce the computational load, the lower boundary is a fixed boundary used to support the overlying rock layers, and the right boundary is a roller support used to represent the infinitely extended rock layers. The humidity field solution from the previous section is applied. It is better to observe the differences between different strength criteria and flow rules and then further discuss the burial depth. Moreover, in a study by J. Pérez-Romero on the Trasvasur Tunnel (Canary Islands, Spain), it was shown that the expansion plane appears at different relative positions to the cross-section of the tunnel: below the floor, at the height of the sidewalls, or above the roof. In this case, it is particularly recommended to use circular or similar cross-sectional shapes as they are the most effective in absorbing the thrust generated by the expansion plane [31].

First, it is necessary to introduce in situ stress into the soil before tunnel excavation. Subsequently, we can calculate the plastic behavior after removing the soil corresponding to the tunnel. The in situ stress needs to be incorporated into the second step. These operations can be set up directly in COMSOL Multiphysics. We can start by adding a solid step to compute the in situ stress. Then, soil plasticity characteristics are added in the second step and the same computational steps are performed. Finally, we can calculate the results. The average ground stress (unit N/m^2^) cloud map under the action of gravity is obtained, characterized by the von Mises stress, and the following figures are the same.

## 3. Consideration of Moisture Expansion Soft Rock Tunnel Excavation Mechanics Response with Different Parameters

### 3.1. Diverse Strength Criteria and Non-Associated Flow Rules in the Context of Humidity Field Theory

Building upon the humidity field solution delineated in the preceding section, this study postulates an axisymmetric distribution of humidity, incorporating the tenets of humidity stress field theory. This approach is particularly pertinent in the context of expansive soft rock, characterized by a series of interrelated processes: (1) exposure to water precipitates notable alterations in the humidity field of expansive soft rock; (2) these humidity field changes, in turn, induce the volumetric expansion and physical softening of the rock material; (3) such expansion and softening processes lead to consequential modifications in both the stress field and displacement field, manifesting a complex coupled relationship.

The first soil plasticity model was developed based on the Mohr–Coulomb criterion, a generalization of methods for obtaining continuous materials and multi-axial stress states. It is defined as follows: when the combination of shear stress and normal stress on any plane reaches a critical condition, the material starts to yield or even fracture. The critical condition is given by the following equation:(1)τ=c−μσ

Here, *τ* is shear stress, *σ* is normal stress, *c* is cohesion (representing the shear strength when the normal stress is 0), and μ=tanϕ is the coefficient of internal friction from the famous Coulomb friction model. This equation represents two lines in the Mohr plane. The stress state is safe when all three Mohr circles lie between these two lines; however, when one of these circles is a tangent to these two lines, it represents the critical state (material starts to yield).

Mohr circles are based on principal stresses *σ*_1_, *σ*_2_, and *σ*_3_. The stress state is calculated using the formulas $\tau =\frac{1}{2}(\sigma_{1}-\sigma_{3})\cos\phi \;$ and $\sigma =\frac{1}{2}\left(\sigma_{1}+\sigma_{3}\right)+\frac{1}{2}(\sigma_{1}-\sigma_{3})\sin\phi$. Therefore, the yield criterion and Equation (1) can be generalized as follows:(2)$$f_{y}\left(\sigma \right)=\left(\sigma_{1}-\sigma_{3}\right)+\left(\sigma_{1}+\sigma_{3}\right)\sin\phi -2c\cos\phi$$

This can even be seen as a specific case of a more general series of criteria based on Coulomb friction, expressed as an equation based on the invariants of the stress tensor:(3)$$f_{y}\left(\sigma \right)=F\left(J_{2},J_{3}\right)+\lambda I_{1}-\beta$$

The Mohr–Coulomb criteria define hexagonal pyramids in the principal stress space, providing great convenience for the direct analysis of this criterion. However, due to the presence of sharp angles (e.g., where the normal to the yield surface is undefined at sharp edges), dealing with constitutive equations from a numerical calculation perspective is challenging.

Another yield criterion—the Drucker–Prager yield criterion, used to avoid issues related to sharp angles—was developed by modifying the von Mises yield criterion. This criterion not only considers Coulomb friction but also establishes a relationship with hydrostatic pressure:(4)$$f_{y}\left(\sigma \right)=\sqrt{J_{2}}+\alpha I_{1}-k$$

This equation represents smooth cones in the principal stress plane rather than hexagonal pyramids. If the coefficients α and k match those in the Mohr–Coulomb criterion, they are as follows:(5)$$\alpha =\frac{2}{\sqrt{3}}\frac{\sin\phi }{3\pm \sin\phi };k=\frac{2\sqrt{3}\sin\phi }{3\pm \sin\phi }$$

The Drucker–Prager yield surface passes through the interior or exterior vertices of the Mohr–Coulomb hexagonal pyramid, depending on whether the ± sign is positive or negative. Plastic flow direction originates from the so-called ‘plastic potential’, which can be either related or unrelated to plasticity, leading to yielding phenomena (yield function). Based on this, various non-associated flow rules can be developed.

In the Drucker–Prager model, volume plastic flow can be made non-zero using associated rules. Consequently, volume changes occur under applied pressure. However, this behavior contradicts that of many soil materials, especially granular materials. Conversely, non-associated flow rules can be used in cases where plastic behavior is isochoric (volume-conserving), better reflecting the plastic behavior of granular materials.

In the modeling environment of COMSOL Multiphysics, the emphasis is placed exclusively on the phenomena of moisture absorption and consequent expansion. Notably, the model abstains from incorporating the decreased rheological stress attributable to physical softening. Nonetheless, to comprehensively capture the rock material’s elastoplastic response, various strength criteria and non-associated flow rules are integrated into the analysis. We consider the Mohr–Coulomb criterion and Drucker–Prager criterion applicable to rocks, as well as flow rules that are related or unrelated to them.

#### 3.1.1. Elastoplastic Response under the Mohr–Coulomb Criterion

Given the elastoplastic characteristics inherent in rock and soil materials, the Mohr–Coulomb model is employed as the primary framework for initial observations. This model, recognized for its aptitude in simulating the behavior of geotechnical materials, serves as a fundamental basis for analyzing the stress and deformation patterns post-excavation. Utilizing the Mohr–Coulomb strength criterion, coupled with the associated flow rule, the study yields detailed visual representations in the form of stress distribution maps and plastic deformation zone maps (Figure 1). These maps are instrumental in elucidating the changes in the stress state and the extent of plastic deformation within the rock mass following excavation activities. The insights gleaned from these visualizations not only enhance the understanding of the rock’s mechanical behavior under excavation-induced stresses but also aid in the formulation of more effective excavation strategies and support systems in expansive soft-rock environments.

The detailed analysis of stress distribution and deformation patterns in the vicinity of the tunnel presents a nuanced understanding of the geomechanical responses induced by excavation activities (Figure 2). Notably, the maximum stress value recorded is 992.6 KPa, precisely located at the coordinates (0.758, −27.276) beneath the tunnel arch. This region, characterized by elevated stress levels, also exhibits a concentration of plastic deformation zones predominantly at the top and bottom of the tunnel arch, signifying critical areas of structural concern.

The excavation of underground structures invariably leads to a release of stress on the local ground surface. This stress release manifests as displacement and uplift, fundamentally altering the ground’s mechanical equilibrium. The phenomenon is further intensified by expansive stress, a critical factor in the geotechnical analysis of tunneling. Expansive stress provokes a volumetric expansion of the surrounding rocks, culminating in notable surface uplift. This uplift not only impacts the stability of underground structures but also triggers observable horizontal ground displacement and surface subsidence (Figure 3).

A meticulous examination reveals that the maximum horizontal ground displacement attains a value of −0.686 mm, approximated to three decimal places, and is observed at 23.4 m from the tunnel axis. Intriguingly, a horizontal displacement of 0.005 mm in the positive *x*-axis direction is detected at a mere 1.8 m from the tunnel axis, presumably attributable to the uplift of the tunnel arch. The displacement profile exhibits a characteristic U-shaped curve as the distance from the tunnel axis increases, indicative of the surrounding rock displacement under the influence of expansive stress. Ground uplift reaches its zenith at 27.654 mm, precisely at the origin of the *x*-axis. This value gradually diminishes with increasing distance from the tunnel axis, registering a minimum of 26.565 mm at 21.96 m away. Beyond this point, a slight increase is noted, likely due to the marginal extrusion of the farther rock mass under expansive stress. The differential between the maximum and minimum uplift values is relatively modest, spanning only 1.089 mm. However, it is significantly larger than the horizontal displacement of the ground, indicating that the predominant mode of ground displacement is uplift.

The mechanical response to tunnel excavation presents a contrasting picture in a scenario devoid of expansive stress considerations (Figure 4). The peak stress value under such conditions is observed to be 787.581 KPa, located at coordinates (5.933, −20.894) below the tunnel arch. In this instance, high-stress and plastic zones are distinctly concentrated along the sides of the tunnel, highlighting the differential impact of expansive stress on underground structure stability.

#### 3.1.2. Mohr–Coulomb Criterion and Non-Associated Flow Rule

In the realm of geomechanical modeling, particularly when addressing the stress–strain behavior of geomaterials, the associated flow rule often falls short of accurately representing the actual material response. Consequently, implementing non-associated flow rules has become prevalent in advanced geotechnical analyses. These rules permit a nonlinear correlation between shear stress and strain, adeptly capturing the intricate nature of geomaterials. Compared to their counterparts, non-associated flow rules offer a more precise prediction of deformation and failure modes of geomaterials under varied stress conditions. Despite their significance, the literature often lacks explicit references to specific non-associated flow rules, primarily because their selection hinges on several factors such as geomaterial type, stress path, and empirical data. This leads to divergent choices of non-associated flow rules across different research domains and practical applications.

A frequently employed non-associated flow rule is the Drucker–Prager criterion, which extends the Mohr–Coulomb criterion by incorporating modification factors to better represent the nonlinear flow behavior of geomaterials. Other notable non-associated flow rules include the Cam-Clay and Cap-Model criteria, each offering unique applicability under specific geotechnical scenarios.

This section endeavors to elucidate and compare the efficacy of prominent strength criteria and flow rules in characterizing geomaterials. Initial observations are drawn using the Mohr–Coulomb criterion in conjunction with the Drucker–Prager matching compression meridian, followed by an application of the Mohr–Coulomb criterion with the Drucker–Prager matching tensile meridian. The outcomes of these applications are visually represented through stress distribution maps and plastic deformation zone maps post-excavation, as depicted in Figure 5.

In the application of the Mohr–Coulomb criterion coupled with the Drucker–Prager matching compressive meridian, our analysis reveals a peak stress occurrence at the coordinates (0.758, −27.276), situated just beneath the tunnel crown. This stress is quantified at a substantial 995.371 KPa. In a comparative vein, the utilization of the Mohr–Coulomb criterion in tandem with the Drucker–Prager matching tensile meridian identifies the maximum stress at the identical location. However, the stress intensity here is marginally reduced to 992.831 KPa, underscoring a nuanced differential impact attributable to the compression versus tensile meridian of the Drucker–Prager model.

A significant observation from these simulations is the concentration of high-stress and plastic deformation zones predominantly at the apex and base of the tunnel arch. This spatial distribution remains consistently pronounced under both flow rules. Intriguingly, the choice between the compressive and tensile meridians of the Drucker–Prager criterion appears to exert a minimal influence on the overall stress distribution post-excavation and the resultant development of plastic zones within the geomaterials. This finding suggests a certain level of robustness in the stress response of the tunnel structure to the applied flow rules. A methodological approach involves subtracting the respective plastic-zone cloud maps to delve deeper into the comparative analysis of plastic zone development under these two distinct flow rules. This process, aimed at highlighting the differential development of plastic zones, is meticulously carried out and the results are encapsulated in the subsequent figure.

In the nuanced realm of geotechnical modeling, applying the Mohr–Coulomb criterion in conjunction with the Drucker–Prager matching compression meridian reveals a distinctive pattern in the development of the plastic zone. This pattern is prominently observed near the arch shoulder and foot of the tunnel, indicative of critical stress concentrations in these areas. Conversely, when the Mohr–Coulomb criterion is paired with the Drucker–Prager matching tension meridian, the plastic zone exhibits more pronounced development beneath the tunnel’s bottom arch, suggesting a different stress distribution characteristic. Further examination of the horizontal displacement and surface uplift reveals that the response curves derived from these two non-associated flow rules align closely with those obtained using the associated flow rule. This similarity underscores the underlying mechanical consistency of the rock mass under varying flow rule scenarios. To clarify the differences, attention is focused on the curve at the maximum value, as comprehensively illustrated in Figure 6.

Figure 7 visually captures the comparative analysis of these flow rules. It becomes evident that the application of non-associated flow rules imposes a constraint on the expansion of expansive soft rock, leading to a marked decrease in both overall horizontal displacement and surface uplift. Notably, the constraint effect is more pronounced when the flow rule matches the compression meridian as opposed to the tension meridian. Specifically, when matching the tension meridian, the results nearly mirror those obtained using the Mohr–Coulomb criterion with the associated flow rule.

Quantitatively, at the point of maximum horizontal displacement, there is a reduction of 0.12 µm under the compression meridian flow rule, compared to a modest 0.037 µm under the tension meridian. This indicates a 0.12‰ more significant reduction with the compression meridian. Similarly, for maximum surface uplift, the compression meridian results in a reduction of 0.4 µm while the tension meridian sees a minor reduction of 0.037 µm. Although these values are relatively small, especially in the context of the scale of surface uplift, they are significant in understanding the nuanced behavior of expansive soft rock under different stress conditions. Subsequently, horizontal displacement and surface uplift patterns derived from these two non-associated flow rules are examined, as shown in Figure 7. The overall trends align closely with those observed under the associated flow rule, underscoring the subtle yet significant differences brought forth by the non-associated rules. Notably, the non-associated flow rules impose constraints on the expansion of expansive soft rock, leading to a diminution in both horizontal displacement and surface uplift. The constraint imposed by the compression meridian is observed to be more effective than that by the tension meridian. In the context of horizontal displacement, a reduction of 0.12 µm is noted with the compression meridian, compared to a 0.037 µm reduction with the tension meridian. Similarly, in terms of surface uplift, the compression meridian results in a reduction of 0.4 µm while the tension meridian shows a marginal reduction of 0.037 µm.

#### 3.1.3. Drucker–Prager Model

In this study, a Drucker–Prager model is established for comparative analysis with three other models. The resulting stress distribution maps, plastic deformation zone maps, surface horizontal displacement curves, and surface uplift curves demonstrate a high degree of similarity to those obtained from the previously mentioned models. Nonetheless, to identify subtle variances, the focus is shifted to the peak values of surface horizontal displacement and surface uplift. The analysis reveals that the rock in the Mohr–Coulomb model exhibits greater hardness than that in the Drucker–Prager model, leading to more significant expansion in the latter. When the Mohr–Coulomb strength criterion is applied, there are observed increases in horizontal displacement and surface uplift by 0.333 µm and 0.857 µm, respectively, compared to when the Drucker–Prager criterion is applied. This increase is more pronounced in horizontal displacement than in surface uplift, particularly in scenarios of severe expansion. This outcome aligns with the findings from the comparison of two different flow rules although the magnitude of these changes is relatively modest on the overall scale. To gain a more comprehensive understanding, an alternative analytical approach is employed, focusing on the total displacement along the semicircular tunnel. This approach traces the displacement from the center of the arch, along the semicircle, to the center of the bottom arch. This method provides a detailed view of the displacement behavior of the surrounding rock under the influence of the four different models. The findings from this approach are systematically presented in Figure 8.

In the realm of geomechanical modeling, the concept that total displacement is a vector is of significant importance. This study presents a detailed analysis of displacement patterns using the Mohr–Coulomb criterion and the Drucker–Prager matching compression meridian (Figure 9). A displacement cloud map generated under these conditions employs a deformation scale factor of approximately 162.73, providing a vivid representation of the geomechanical behavior of the surrounding rock. This visual depiction reveals a compression effect on both sides of the tunnel and an overall uplift of the surrounding rock.

Interestingly, it is observed that except for the specific cases of the Mohr–Coulomb criterion and Drucker–Prager’s matching compression meridian (Figure 10), the displacement patterns of the surrounding rock for the other three criteria exhibit remarkable similarities. However, under the Mohr–Coulomb criterion and Drucker–Prager’s matching compression meridian, a distinct behavior is noted: an increase in the displacement of the surrounding rock at the bottom arch and a decrease at the top arch. This finding corroborates the earlier identified differences in the development of the plastic zone.

The same pattern of changes is also evident in the application of the Mohr–Coulomb criterion and Drucker–Prager’s matching tension meridian. This trend underscores the fact that restricting expansion leads to a reduction in displacement at the top arch and an increase at the bottom arch. When compared with the Mohr–Coulomb criterion and the associated flow rule, the displacement of the surrounding rock under the Drucker–Prager criterion’s matching compression meridian is observed to increase by 0.465 mm at the bottom arch and decrease by 0.445 mm at the top arch. This variation indicates a notable impact on the displacement behavior of the surrounding rock, contributing to a better understanding of the mechanical response of rock masses to tunnel excavation.

Li [32] conducted on-site tunnel excavation monitoring and numerical simulations on the conglomerate section of the three-linked Sixzhan Complex Line Tunnel between the BeiKaizhu and Qiewu stations in Yunnan Province, from D1K305+945 to D1K306+440. On site, it was found that after securing the support of the level guide and the connecting channel, initial support cracking, arch distortion, and the phenomenon of bottom plate (already built with a reverse arch) lifting occurred at multiple locations. Furthermore, the deformation of the sidewalls on both sides was more severe than that of the arch. The heavily deformed section of the middle-level guide was from PDK305+945 to PDK306+273, where the grid arch support was used, and deformation cracks, arch top, and sidewall cracks occurred soon after the spacing construction, resulting in uplift at the tunnel bottom. The numerical simulation showed that the maximum vertical displacement occurred at the tunnel arch’s top and bottom, with the arch top showing sinking deformation and the bottom showing uplift deformation. The maximum horizontal displacement occurred near the arch foot. Overall, the maximum deformation of the tunnel wall occurred at the tunnel bottom while the minimum deformation occurred near the arch top.

### 3.2. Shear Dilation

In the context of tunnel excavation, the phenomenon of shear dilation is an inherent and critical aspect of underground engineering. Shear dilation, in geotechnical terms, refers to the process where horizontal stresses exerted by the surrounding soil or rock lead to an increase in vertical stress. This phenomenon often results in a non-uniform distribution of stress within the surrounding rock, particularly in areas subjected to high stress. When shear dilation is considered, the stress–strain curve of the surrounding rock more accurately depicts this non-uniformity, offering a clearer understanding of the geomechanical behavior of the rock mass.

In this study, the focus is placed on understanding the implications of varying shear dilation angles within the framework of the Mohr–Coulomb criterion and the Drucker–Prager matching-compression-meridian model. The shear dilation angle is the arctangent value of the ratio of normal displacement to tangential displacement that occurs on structural surfaces of rock during shear deformation. Previous comparisons have been made under different strength criteria and associated or non-associated flow rules. Therefore, this analysis examines, explicitly, the changes brought about by different shear dilation angles. The shear dilation angles are set at 0°, 5°, 10°, and 20°, and the resulting stress distribution maps, plastic deformation zone maps, surface horizontal displacement, and surface uplift under the Mohr–Coulomb criterion with the Drucker–Prager matching compression meridian are analyzed. The outcomes are found to be broadly consistent with those of the model without shear dilation considered. The maximum stress consistently occurs at the point (0.758, −27.276), aligning with the maximum stress value observed in the model devoid of shear dilation. As the shear dilation angle increases, the maximum stress values recorded are 987.575 KPa, 984.436 KPa, 983.524 KPa, and 983.128 KPa. This trend indicates a decrease in maximum stress with an increasing shear dilation angle although the rate of decrease slows down progressively.

Furthermore, the plastic zone remains unchanged, maintaining the patterns observed in previous models. The study continues to focus on the differences in the peak values of surface horizontal displacement and surface uplift, which are critical indicators of the tunnel’s stability and the surrounding rock’s response to excavation processes (Figure 11).

Analysis reveals that considering dilation leads to an expansion and displacement of the surrounding rock. When the ground surface angle (*Ψ*) is set to 0°, the maximum horizontal displacement and maximum surface uplift increase by 0.345 μm and 1.129 μm, respectively, compared to scenarios without dilation. These increases correspond to growth rates of 0.503‰ for horizontal displacement and 0.0408‰ for surface uplift. This finding substantiates the hypothesis that as rock expansion intensifies, the horizontal displacement at the ground surface tends to develop more significantly than surface uplift.

Furthermore, when comparing the dilation angle of *Ψ* = 0° with *Ψ* = 20°, the maximum horizontal displacement and maximum surface uplift exhibit increases of 0.101 μm and 0.157 μm, respectively. These values correspond to growth rates of 0.147‰ for horizontal displacement and a notably lower rate of 5.7 × 10^−6^ for surface uplift. The analysis of displacement curves under various dilation angles reveals a decelerating trend in displacement growth as the dilation angle increases. To comprehensively understand the impact of dilation on tunnel excavation, the study evaluates the maximum horizontal displacement and maximum surface uplift of the ground at different dilation angles. This evaluation takes into consideration both the Mohr–Coulomb criterion with the Drucker–Prager matching compression meridian and the Mohr–Coulomb criterion with the Drucker–Prager matching extension meridian (Figure 12).

In geotechnical modeling, it is noted that when the dilation angle is minimal, the maximum horizontal displacement and maximum surface uplift values under different non-associated flow rules tend to be similar, often closely matching each other. However, with an increase in the dilation angle, the discrepancy becomes more pronounced. In these cases, the matching compression meridian exhibits a notable effect in restricting expansion although the difference in comparison to other flow rules is marginally lower. Following this observation, the analysis then shifts to examining the displacement of the surrounding rock, maintaining the methodology previously established (Figure 13).

In the domain of geotechnical analysis, it has been established that an increase in rock expansion influences the displacement patterns of the surrounding rock in tunnel excavations. Specifically, a rise in expansion leads to an increase in the displacement of the top arch of the tunnel while resulting in a decrease at the bottom arch. When evaluating the displacement curves at different dilation angles, it is observed that they tend to align closely. Therefore, the analysis is concentrated on the displacement of the surrounding rock precisely at the center of the tunnel’s top and bottom arches under various dilation angles and considering both the Mohr–Coulomb criterion and the Drucker–Prager matching extension meridian (Figure 14).

As the dilation angle escalates, a discernible trend is observed: the displacement at the top arch diminishes whereas the displacement at the bottom arch exhibits an increase. This pattern holds true across the range of dilation angles considered. When the two different flow rules are compared, particularly under the matching compression meridian, it is noted that the increase in dilation angle leads to relatively smaller displacement at the top arch and larger displacement at the bottom arch, as opposed to the outcomes observed under the matching tension meridian.

### 3.3. Buried Depth 

Following the comparative analyses conducted, a specific model configuration is selected: a dilation angle of *Ψ* = 0° with the application of the Mohr–Coulomb criterion and the Drucker–Prager matching compression meridian. To further explore the impact of different model sizes and burial depths, the dimensions of the model are adjusted to 45 m × 55 m, 45 m × 65 m, and 45 m × 75 m, each retaining the same parameters as in the initial setup. These alterations facilitate the creation of models with varying buried depths of 30 m, 40 m, and 50 m. The resulting stress distribution maps and plastic deformation zone maps for these configurations are presented below (Figure 15), offering a visual insight into the stress and plastic zones under these varied conditions.

In the realm of underground construction and geotechnical analysis, the influence of buried depth on stress distribution and deformation patterns is of paramount importance. This study reveals that as the buried depth of the tunnel increases, there is a noticeable outward shift in the location of maximum stress values. These maximum stress points are distinctly identified at coordinates (0, −37.629), (1.066, −47.859), and (3.057, −57.619) for the respective increasing depths. Accompanying this shift is a notable evolution in the high-stress and plastic deformation zones. As the depth increases, both these zones progressively develop into circular forms. This observation suggests that the plastic zone will likely form a near-perfect envelope circle during deep tunnel excavation, encapsulating the area around the tunnel.

The study then turns its focus to examining the maximum horizontal displacement and maximum surface uplift of the ground. This analysis is crucial in understanding the surface-level impacts of deep tunnel excavation. The results of this investigation are systematically presented in Figure 16.

As the burial depth of underground structures increases, a notable trend is observed in the horizontal displacement of the ground surface, which tends to decrease, and the corresponding displacement curve exhibits a smoother profile. This phenomenon can be attributed to the fact that with an increasing depth, the surrounding rock is subjected to more significant stress. However, the deformation capacity of the rock is inherently limited by its finite strength, resulting in a reduction in horizontal displacement at the ground surface.

In addition to changes in horizontal displacement, variations in the uplift phenomenon at the ground surface are also evident with varying burial depths. At shallower depths, the uplift of the ground surface is primarily characterized by the bulging effect at the top arch, which results from the compression of the surrounding rock in this area. Conversely, with an increasing burial depth, the compression affects rock masses further away from the top arch. This shift causes the center of the ground surface uplift to move outward from the top arch, and the overall magnitude of the uplift gradually intensifies.

This study then proceeds to analyze the displacement of the surrounding rock in relation to different burial depths. This analysis is particularly focused on a semicircular path, taken as the *x*-axis, beginning from the center of the top arch and extending to the center of the bottom arch. The findings from this investigation are comprehensively illustrated in Figure 17.

With the progressive increase in burial depth, the surrounding rock is subjected to escalating levels of stress. However, given the inherent limitations in the strength of these geological materials, they are unable to withstand excessive stress, ultimately leading to localized failure in certain areas. This phenomenon is particularly evident at the bottom arch of the tunnel, where the impact of greater compression forces becomes pronounced, manifesting as significant displacement. In contrast, the displacement at the top arch is considerably restricted due to the differing stress conditions. This pattern of deformation highlights a critical aspect of deep burial conditions in tunnel engineering: the deformation of the surrounding rock is predominantly concentrated at the bottom arch. In such scenarios, the rock masses at the lower sections of the tunnel exhibit more notable responses to the applied stresses, resulting in greater displacement and potential deformation. Conversely, the rock at the top arch, being subjected to less intense stress, remains relatively stable, exhibiting minimal displacement. With the increase in burial depth, the distribution of surrounding rocks varies, leading to different issues of prominence at the arch crown and arch bottom during the excavation process. For example, tunnels such as Enasan Tunnel [33], Mugua Tunnel [34], Jiazhuqing Tunnel [35], Muzhaoling Tunnel [36], and Wushaoling Tunnel [37] all experienced significant arch crown settlement during the construction phase. Similarly, tunnels like Cuijiagou Tunnel [38] and Guanjiao Tunnel [39] experienced bottom uplift and bottom heave during both the construction and operation phases.

### 3.4. Support Resistance

We choose the model with a burial depth of 50 m, with the overlying rock pressure *P*_0_ as the reference, to consider different support resistances *P*_i_ = 0.10$P_{0}$, *P*_i_ = 0.15*P*_0_, *P*_i_ = 0.20*P*_0_, and *P*_i_ = 0.25*P*_0_ and obtain stress cloud maps and plastic-zone cloud maps, which are shown in Figure 18.

In this study, a targeted model with a burial depth of 50 meters is selected to explore the effects of varying support resistances on the geomechanical behavior of the surrounding rock. The overlying rock pressure, denoted as *P*_0_, serves as a reference point for quantifying different levels of support resistance. These levels are methodically varied as *P*_i_ = 0.10*P*_0_, *P*_i_ = 0.15*P*_0_, *P*_i_ = 0.20*P*_0_, and *P*_i_ = 0.25*P*_0_ to assess their impact on the stress distribution and plastic deformation characteristics of the rock mass.

The outcomes of these variations in support resistance are meticulously captured through stress distribution maps and plastic deformation zone maps, as depicted in Figure 18. A notable observation from these maps is the formation of a high-stress zone, characterized by its circular shape. This zone is observed to gradually shift outward as the level of support resistance increases. Alongside this, the plastic deformation zone also assumes a circular configuration, exhibiting progressive development in response to the escalating levels of support resistance.

Moreover, the study conducts a detailed observation of the stress patterns, tracing a path from the center of the tunnel and extending along the positive direction of the *x*-axis. This directional analysis is crucial in gaining insights into how stress propagates through the rock mass in response to the applied support resistance. The results of this observation, offering a comprehensive view of the stress distribution under different support resistance scenarios, are presented in Figure 18.

As the support resistance increases, the stress around the tunnel decreases. However, when the support resistance increases again, a negative stress zone appears, leading to an error. Figure 19 is roughly consistent with the theoretical curve mentioned earlier.

As support resistance intensifies, there is a notable reduction in stress surrounding the tunnel. However, a further increase in support resistance paradoxically gives rise to a negative stress zone, which manifests as an anomaly in the stress distribution. This occurrence is depicted in Figure 20, aligning closely with the previously discussed theoretical curve, underscoring the intricate relationship between support resistance and stress behavior in tunnel environments.

In the analysis of the displacement behavior of the surrounding rock during tunnel excavation, it is observed that the introduction of support resistance leads to a nuanced response in the displacement patterns. Relative to the unsupported condition, the presence of support resistance initially causes a decrease in displacement at the tunnel vault, followed by an increase as the support resistance intensifies. This dynamic indicates that while support resistance effectively restricts displacement at the tunnel vault, the displacement at the tunnel invert conversely escalates with increased support resistance. This pattern reflects the interplay between the effects of support resistance and expansion stress.

The primary function of the support structure in tunnel construction is to stabilize the rock surrounding the tunnel and mitigate displacement and deformation. When support is in place, the structure absorbs a part of the loads at the tunnel vault, which in turn moderates the fluctuation of displacement in this area. However, at the tunnel invert, where substantial vertical loads are present, the effectiveness of the support structure is comparatively limited, resulting in relatively more significant displacement. Zhang et al. (2024) [40] conducted a detailed study on the exit section of a dual-track high-speed railway tunnel located in Northern China (DK35+190~DK36+609, length 1419 m). The test mudstone clay had a high mineral content of 68.8% and was buried at a depth of over 100 m. Both on-site testing and numerical simulations showed that with support, the maximum settlement occurred at the arch crown.

To comprehensively understand the factors driving the observed increases and decreases in displacement, it is imperative to consider the mechanical properties of the surrounding rock, the stiffness of the support structure, and the material properties of both. In conditions of low support resistance, the rock at both the top and bottom of the tunnel is subjected to significant deformation forces, culminating in increased displacement. Conversely, as support resistance escalates, the support structure becomes more capable of constraining the deformation of the surrounding rock, thus moderating the extent of displacement variation (Figure 21).

### 3.5. Case Verification and Analysis

#### 3.5.1. Case Verification

In this study, a deeply buried tunnel, surrounded by expansive rock, is analyzed under a static-water-pressure environment of *P*_0_ = 1.5 MPa, with the support resistance set at *P*_i_ = 0.16 MPa. All additional parameters conform to those specified in Table 1. The analysis employs the Mohr–Coulomb criterion in conjunction with the Drucker–Prager matching compression meridian, maintaining a dilation angle (*Ψ*) of 0°. The investigation includes the generation of stress distribution maps and plastic deformation zone maps, both considering and not considering the effects of expansion stress and shear dilation. These maps, which provide a detailed visual representation of the stress and deformation states, are showcased in Figure 22.

In the absence of expansion stress, the high-stress zone and plastic zone predominantly localize at the sides of the tunnel. Conversely, when considering both expansion stress and shear dilation, these zones evolve into circular patterns and exhibit outward development from the tunnel sides. Stress observations are focused from the tunnel center, extending towards the positive *x*-axis direction.

The consideration of expansion stress and shear dilation results in a surrounding rock stress curve that aligns well with theoretical predictions. In contrast, omitting expansion stress leads to a lower radial stress curve, as the high-stress zone is concentrated at the tunnel sides, and a higher normal stress curve compared to scenarios where expansion and shear dilation are factored in. Figure 23 in this paper visually contrasts the surrounding rock stress curves under conditions with and without consideration of expansion stress and shear dilation.

#### 3.5.2. Rock Damage

During the process of tunnel excavation and rock swelling due to water absorption, rock damage naturally occurs. This study considers rock damage based on a specified case, employing the previously mentioned parameters. The model utilizes a scalar damage approach, introducing a scalar parameter to quantify material damage within the rock. This model follows an exponential strain softening pattern, where material strength diminishes with increasing strain, a common description of the damage evolution in rock materials under strain. The fracture energy per unit area is considered at 1 × 10^6^ J/m^2^.

The stress-distribution and plastic-zone diagrams, incorporating damage considerations, are generated and compared with a model that does not account for damage. These findings are illustrated in Figure 24, visually comparing the damaged and undamaged scenarios.

The analysis of the figures reveals that considering damage in rock leads to a slight expansion of the high-stress concentration zone and a reduction in maximum stress compared to scenarios without damage. This expansion and stress reduction can be attributed to damage-induced weakening of rock strength, which renders previously stress-resistant areas more susceptible. The presence of damage also facilitates partial stress relief, thereby decreasing maximum stress levels.

In contexts such as tunnel and deep-foundation-pit excavation, rock damage plays a critical role. The alteration of rock strength and stiffness due to damage significantly impacts stress distribution within the rock mass. Additionally, damage consideration indicates the potential plastic deformation of the rock, as evidenced by the development of the plastic zone. This suggests a reduction in the rock’s resistance to damage and deformation, leading to adaptation to external stress changes.

However, these observations are based on specific experimental or numerical simulations and are subject to various influencing factors. In practical applications, a comprehensive evaluation of rock damage and stress analysis, tailored to the geological and construction conditions, is essential. This is crucial for developing effective support and reinforcement strategies, ensuring the safety and integrity of underground engineering projects.

For a clear comparative understanding, this paper includes different stress diagrams with and without damage consideration, as shown in Figure 25.

When damage is taken into account, a decrease in stress around tunnel excavation sites is observed, suggesting the potential for rock fractures under high-stress conditions. Notably, stress increases at the center of the tunnel arch, possibly due to varying geological conditions.

These observations align with findings from extensive practical experience and have been corroborated by numerous scholars. For instance, research conducted by the Institute of Geophysics at the Chinese Academy of Sciences has examined the impact of tunnel excavation on surrounding rock stress, resulting in publications on the subject. Their studies confirm that rock stress and crack distribution around tunnels do undergo changes during excavation, which could lead to rock damage or collapse, highlighting the importance of rigorous monitoring and preventive measures in tunnel construction.

The next phase of the study involves examining the horizontal displacement and ground uplift, which are observed to be nearly coincident. These parameters are particularly analyzed at their maximum values to understand their implications more clearly (Figure 26).

When damage is considered, there is a noticeable downward shift in the curves of horizontal displacement and ground uplift, indicating a reduction in ground uplift but an increase in horizontal displacement. This trend could potentially compromise the stability of the rock surrounding the tunnel. The research highlights that the increase in horizontal displacement and the downward shift of the uplift curve are indicative of potential horizontal instability in the surrounding rock. This phenomenon emphasizes the need to carefully monitor surface deformations during tunnel excavation to maintain structural integrity.

The displacement of the surrounding rock in these conditions is illustrated in Figure 27, providing a visual representation of these changes and their potential impact on tunnel stability.

They are in close proximity and nearly overlapping. We measure the displacement at the lowest point of the tunnel arch. When accounting for damage, the displacement measures 23.63399 mm, but without accounting for damage, it measures 23.63389 mm. The displacement at the center of the tunnel arch is 1.84053 mm when considering damage and 1.84083 mm when not considering damage. There is simply a minuscule disparity.

Evidently, the impact of rock damage during tunnel construction is negligible, particularly when there is support resistance. Support measures are crucial for maintaining the stability of the rock surrounding the tunnel. By implementing various support measures such as lining, anchor rods, and arching steel frames, the displacement and deformation of the surrounding rock may be efficiently managed, thereby minimizing the risk of rock damage during tunnel construction. Furthermore, it is essential to choose appropriate support measures according to the particular geological conditions and tunnel configurations in order to guarantee the safety and dependability of tunnel construction and operation.

Furthermore, the influence of rock deterioration on the stress distribution surrounding the tunnel cannot be entirely disregarded. While support systems can partially mitigate the risk of rock damage, stress changes and subsequent rock damage and deformation are nevertheless expected during tunnel excavation. Hence, tunnel engineering necessitates thorough evaluations of the stability of the rock surrounding the tunnel. This assessment should evaluate the attributes and suitability of various support measures in order to provide appropriate construction plans and support designs that ensure the safety and dependability of tunnel construction and operation.

## 4. Conclusions

This study has employed COMSOL numerical simulations to be compared with theoretical numerical solutions based on the theory of the moisture stress field of elastic–plastic systems. This comparison has verified the moisture stress field and affirmed the precision of the hydraulic-coupling mathematical model. This analysis has involved the examination of several strength criteria and non-associated flow rules. The study findings indicate that the Mohr–Coulomb criterion is more efficient than the Drucker–Prager criterion in limiting water-induced swelling in expansive soft rock. More precisely, utilizing the Mohr–Coulomb criterion in conjunction with the Drucker–Prager compression meridian imposes a more stringent limitation on swelling compared to the tension meridian. This limitation results in the decreased lateral movement and upward movement of the earth, accompanied by an increase in movement at the top of the tunnel arch and a relative decrease at the highest point of the tunnel.

(1) Shear dilation substantially impacts the behavior of the rocks in the vicinity. With an increase in the shear dilation angle, the adjacent rock undergoes more significant expansion, resulting in an amplified horizontal displacement and uplift of the earth. The displacement of the adjacent rock diminishes towards the lower part of the tunnel arch and intensifies towards the upper part in accordance with both non-associated flow principles.

(2) The impact of burial depth is significant. As the depth deepens, the horizontal displacement reduces and reaches a stable state, but the ground uplift increases. This shift transitions from a vertical uplift at the highest point of the arch to a lateral expansion towards the remote rock mass. The adjacent rock’s displacement intensifies at the tunnel arch’s lower part and diminishes at the upper part.

(3) The implementation of support resistance results in the creation of a high-pressure area in a circular band, which expands outward as support resistance intensifies. Consequently, the plastic zone gradually transforms into a circular form. Both the horizontal displacement and ground uplift exhibit an increase in magnitude. Additionally, the displacement of the surrounding rock demonstrates an increase at the bottom of the tunnel arch while displaying a decrease at the top when compared to unsupported conditions. This indicates a positive association. 

(4) In the context of tunnel excavation, particularly in moist swelling soft rock tunnels, the influence of rock damage is found to be relatively low.

## Figures and Tables

**Figure 1 materials-17-01747-f001:**
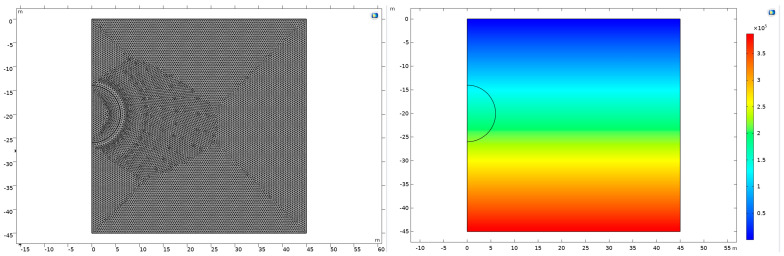
Grid-based model and average ground stress.

**Figure 2 materials-17-01747-f002:**
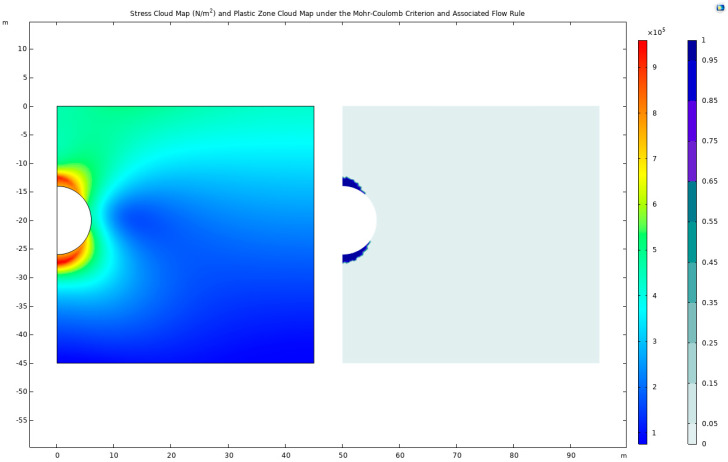
Stress cloud map and plastic-zone cloud map under the Mohr–Coulomb criterion and associated flow rule.

**Figure 3 materials-17-01747-f003:**
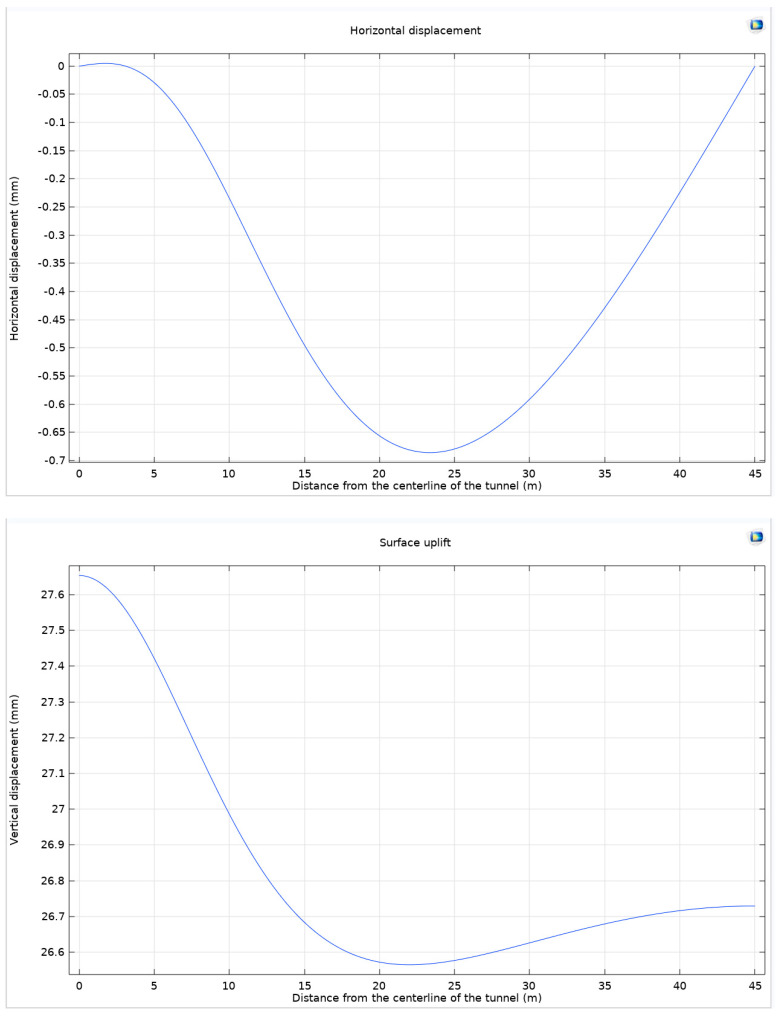
Horizontal displacement of the ground and ground uplift under the Mohr–Coulomb criterion and associated flow rule.

**Figure 4 materials-17-01747-f004:**
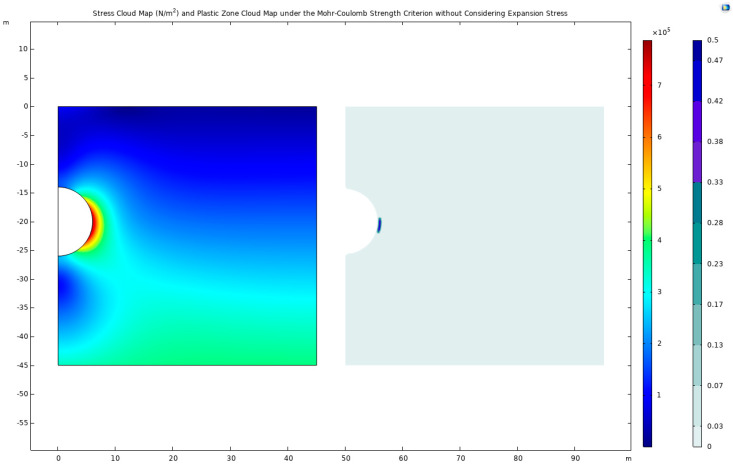
Stress cloud map and plastic-zone cloud map under the Mohr–Coulomb strength criterion without a consideration of expansion stress.

**Figure 5 materials-17-01747-f005:**
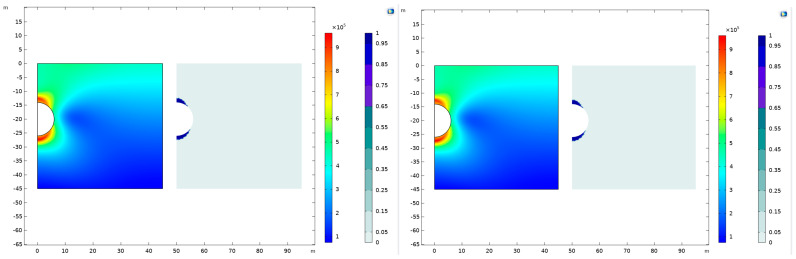
Stress cloud map and plastic-zone cloud map under the Mohr–Coulomb criterion and Drucker–Prager matching compression meridian and Drucker–Prager matching tensile meridian.

**Figure 6 materials-17-01747-f006:**
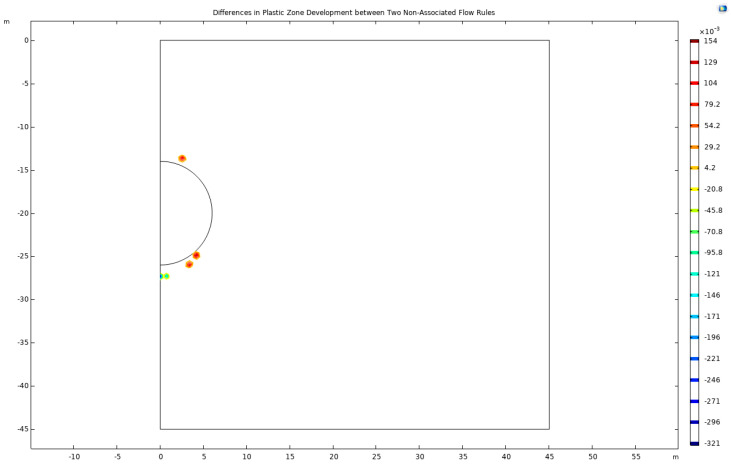
Differences in plastic zone development between two non-associated flow rules.

**Figure 7 materials-17-01747-f007:**
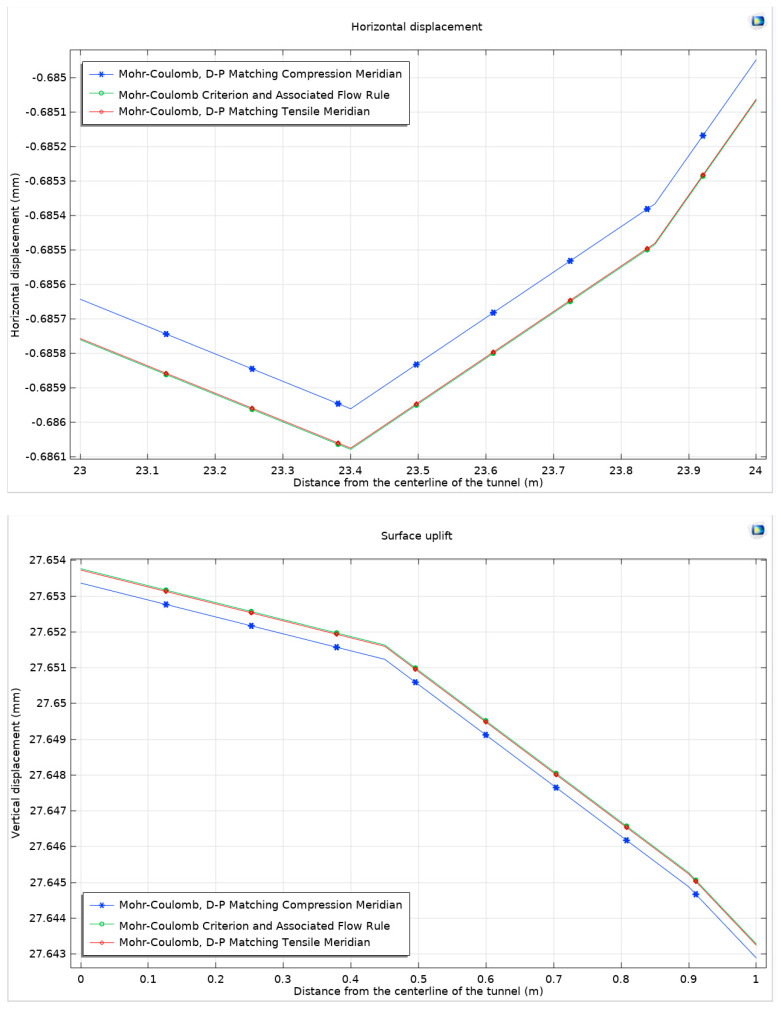
Mohr–Coulomb criterion, surface horizontal displacement, and surface uplift under associated and non-associated flow rules.

**Figure 8 materials-17-01747-f008:**
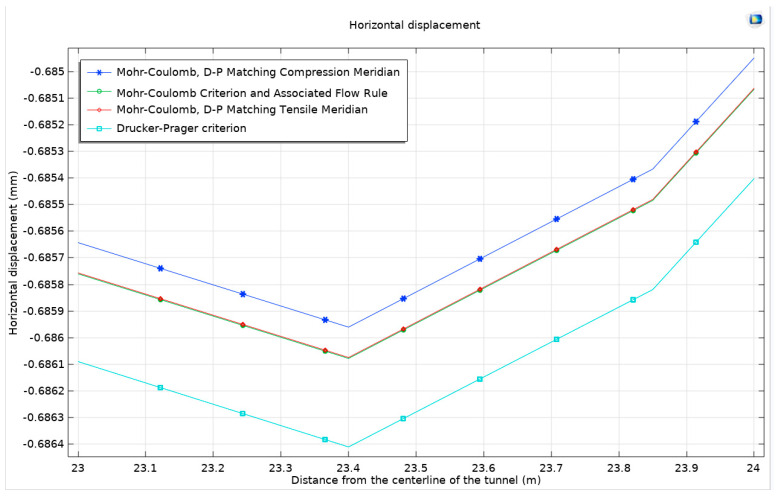
Surface horizontal displacement and surface uplift under different models.

**Figure 9 materials-17-01747-f009:**
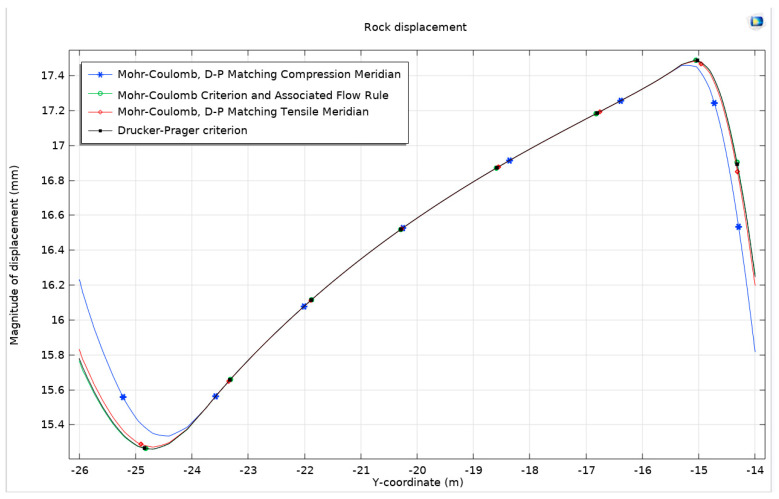
Rock displacement under different models.

**Figure 10 materials-17-01747-f010:**
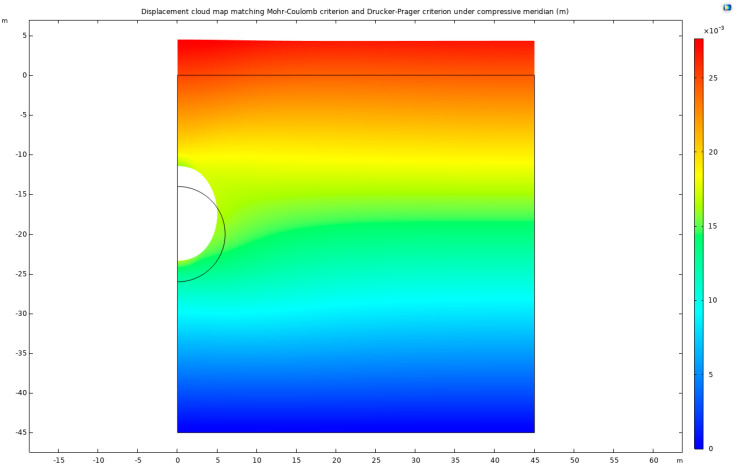
Displacement cloud map matching Mohr–Coulomb criterion and Drucker–Prager criterion under compressive meridian.

**Figure 11 materials-17-01747-f011:**
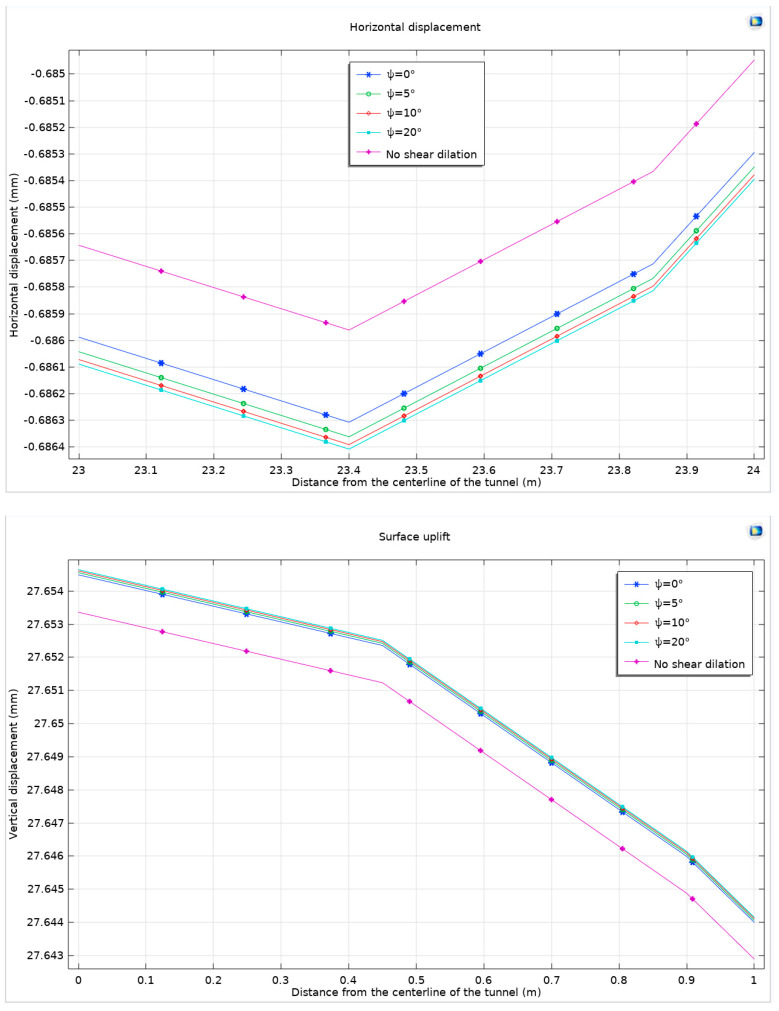
Surface horizontal displacement and surface uplift under compressive meridian matching Mohr–Coulomb criterion and Drucker–Prager criterion considering shear dilation angle.

**Figure 12 materials-17-01747-f012:**
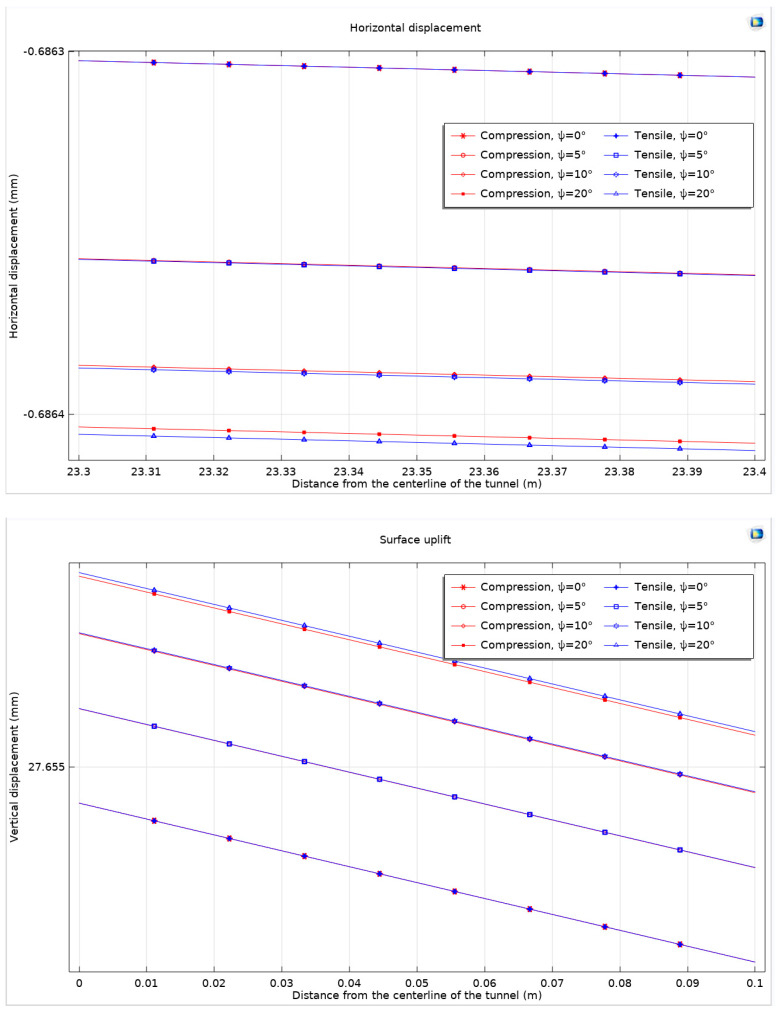
Maximum horizontal displacement and maximum surface uplift considering dilation angle under Mohr–Coulomb criterion and different non-associated flow rules.

**Figure 13 materials-17-01747-f013:**
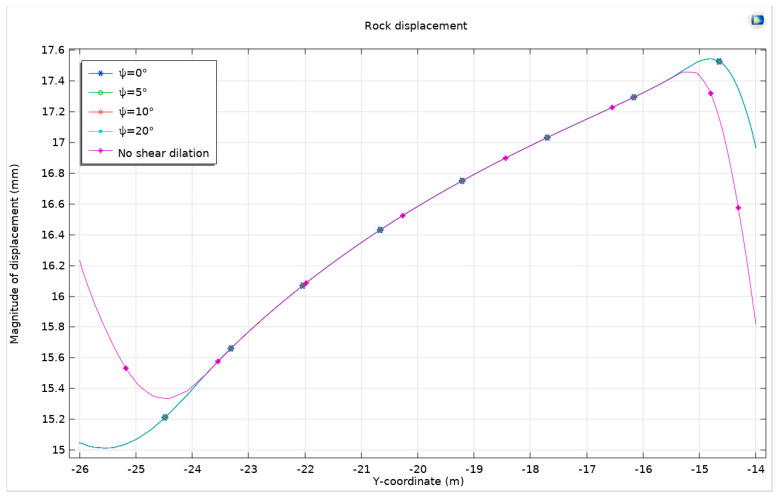
Surrounding rock displacement under Mohr–Coulomb criterion and Drucker–Prager matching compression meridian considering dilation angle.

**Figure 14 materials-17-01747-f014:**
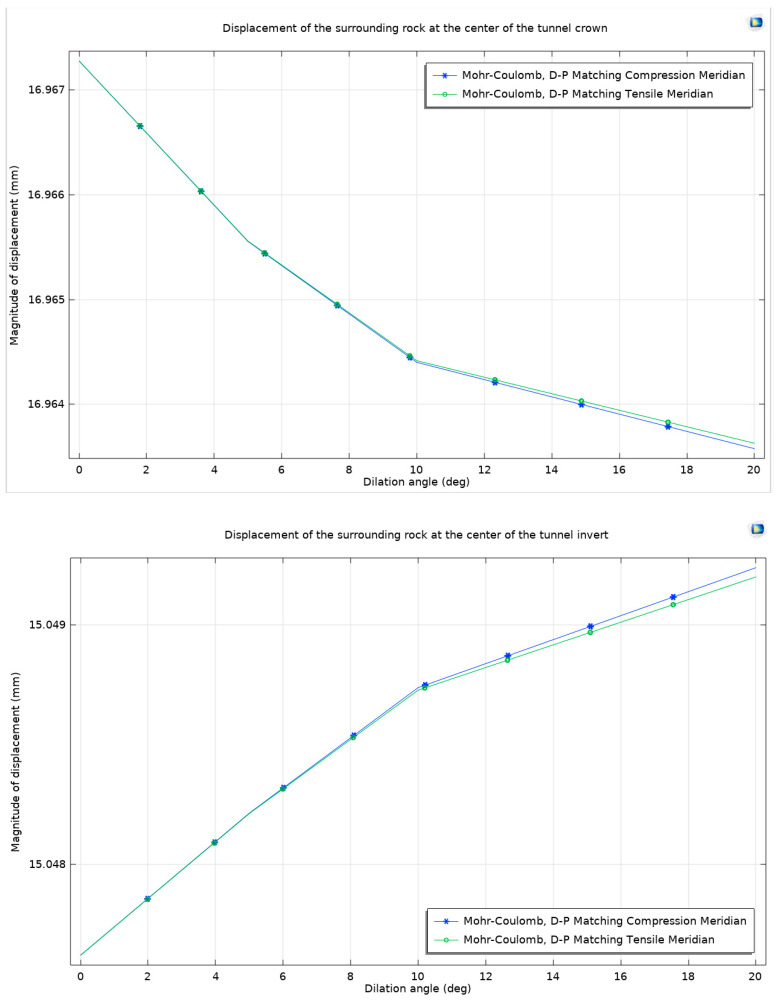
Surrounding rock displacement at the center of the top arch and the bottom arch of the tunnel under Mohr–Coulomb criterion and different non-associated flow rules considering dilation angle.

**Figure 15 materials-17-01747-f015:**
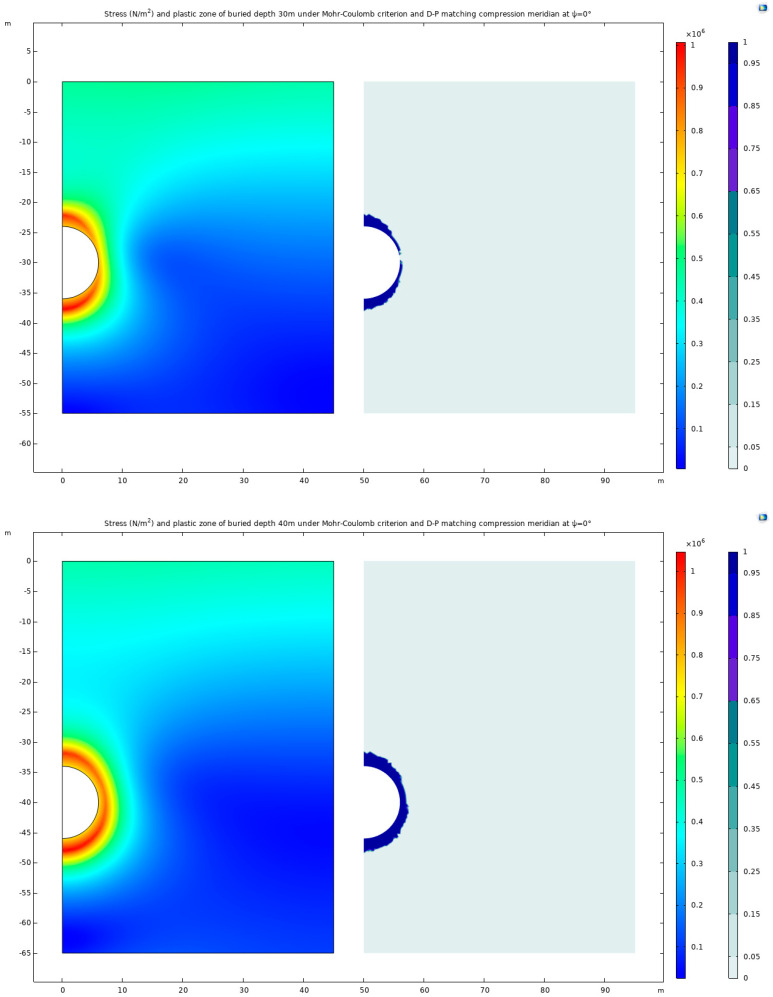
Stress and plastic zone of buried depths 30 m, 40 m, and 50 m under Mohr–Coulomb criterion and Drucker–Prager matching compression meridian at *Ψ* = 0°.

**Figure 16 materials-17-01747-f016:**
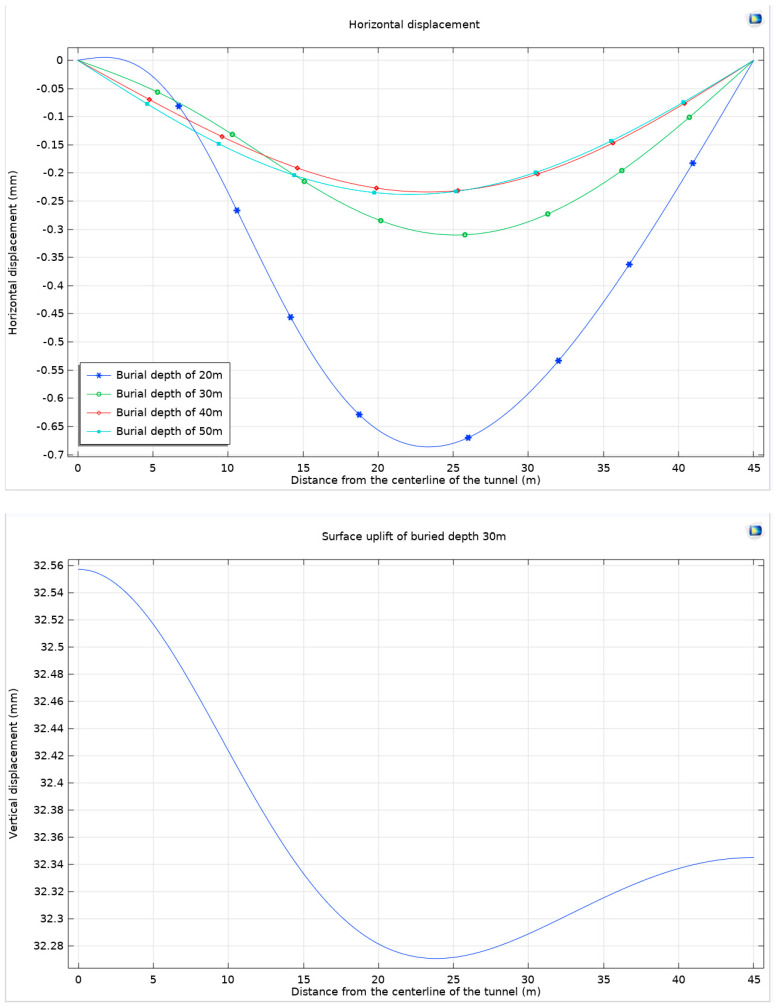
Horizontal displacement and surface uplift of buried depths 30 m, 40 m, and 50 m under Mohr–Coulomb criterion and Drucker–Prager matching compression meridian at *Ψ* = 0° under different buried depths.

**Figure 17 materials-17-01747-f017:**
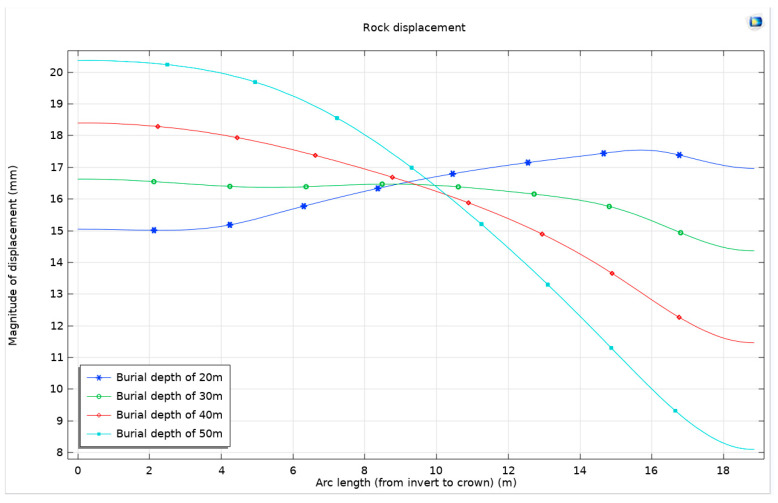
Surrounding rock displacement under Mohr–Coulomb criterion and Drucker–Prager matching compression meridian considering dilation angle at different buried depths.

**Figure 18 materials-17-01747-f018:**
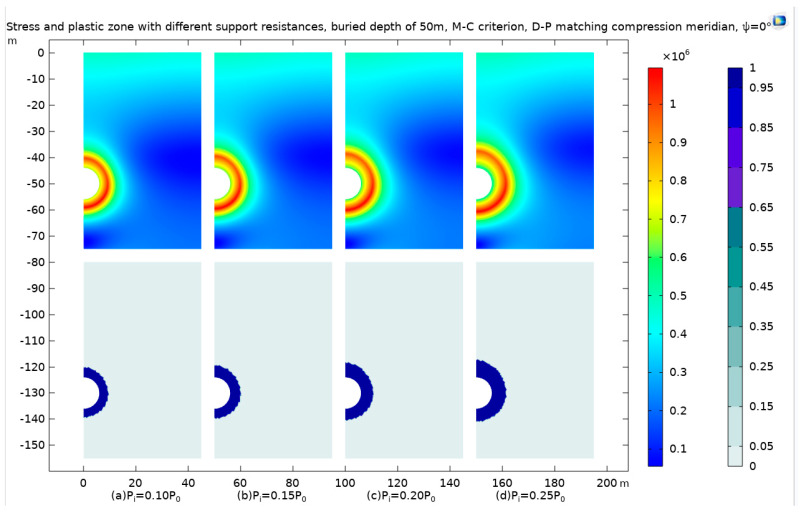
The stress and plastic zone with different support resistances *P*_i_, buried at a depth of 50 m, using the Mohr–Coulomb criterion and Drucker–Prager matching compression meridian with *Ψ* = 0°.

**Figure 19 materials-17-01747-f019:**
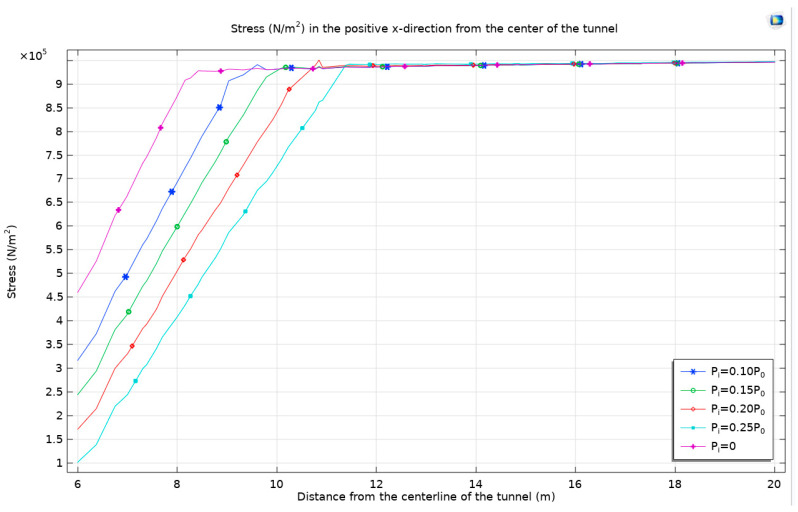
The stress from the tunnel center towards the positive direction of the *x*-axis with different support resistances.

**Figure 20 materials-17-01747-f020:**
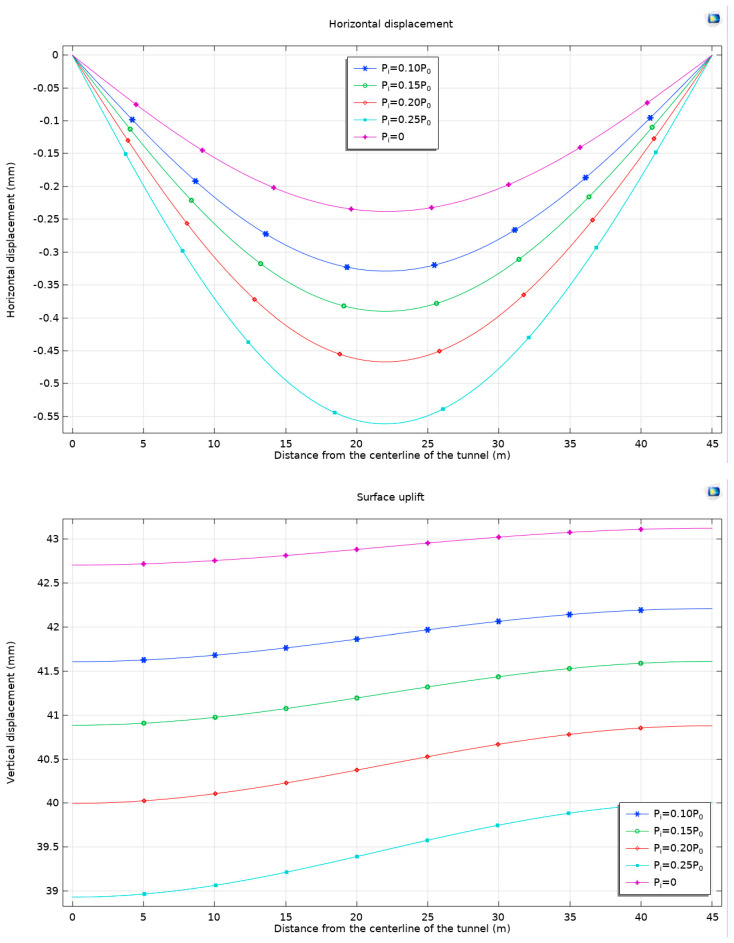
Horizontal displacement and surface uplift with different support resistances *P*_i_, buried at a depth of 50 m, using the Mohr–Coulomb criterion and Drucker–Prager matching compression meridian with *Ψ* = 0°.

**Figure 21 materials-17-01747-f021:**
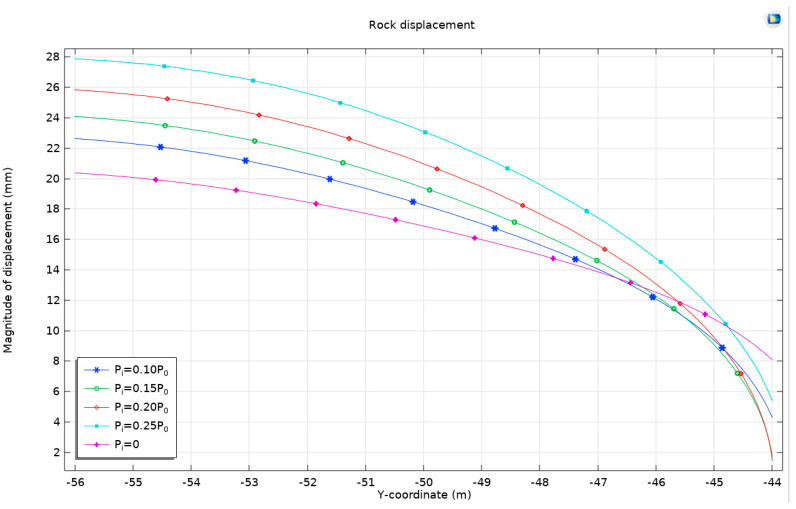
The displacement of the surrounding rock with different support resistances *P*_i_, buried at a depth of 50 m, using the Mohr–Coulomb criterion and Drucker–Prager matching compression meridian with *Ψ* = 0°.

**Figure 22 materials-17-01747-f022:**
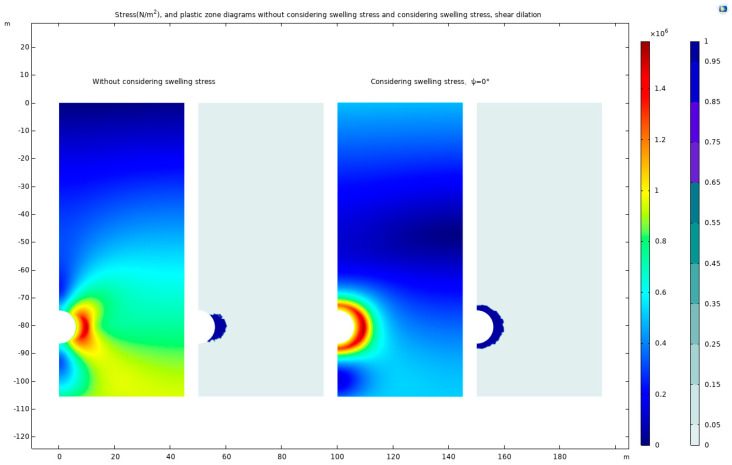
The stress cloud maps and plastic–zone cloud maps, not considering and considering expansion stress and shear dilation.

**Figure 23 materials-17-01747-f023:**
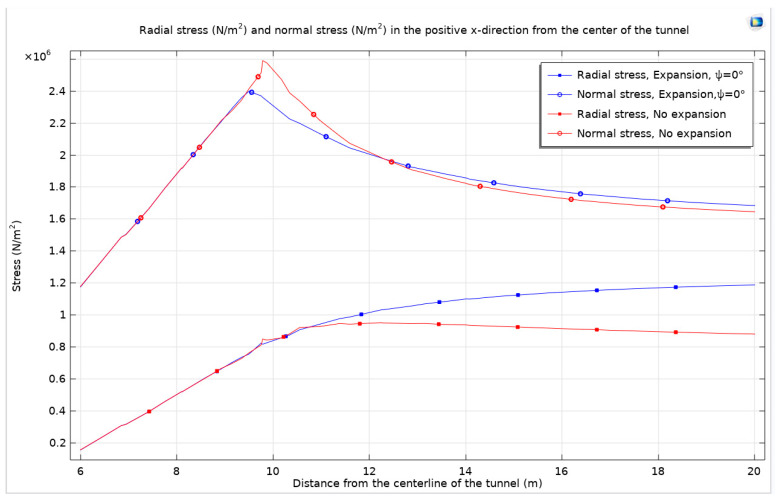
The surrounding rock stress curves, considering and not considering expansion stress and shear dilation.

**Figure 24 materials-17-01747-f024:**
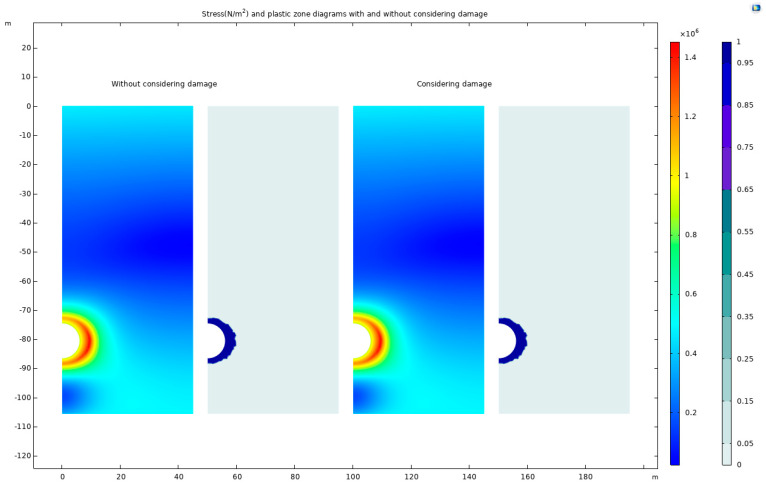
Stress and plastic-zone diagrams without and with a consideration of damage.

**Figure 25 materials-17-01747-f025:**
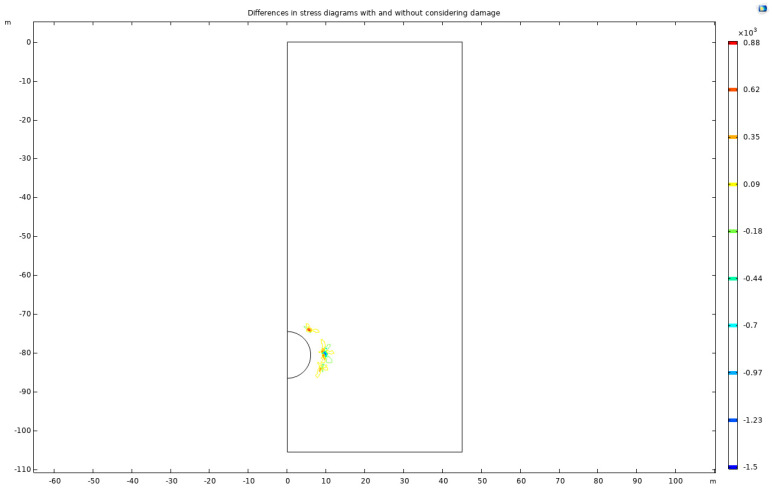
Differences in stress diagrams with and without a consideration of damage.

**Figure 26 materials-17-01747-f026:**
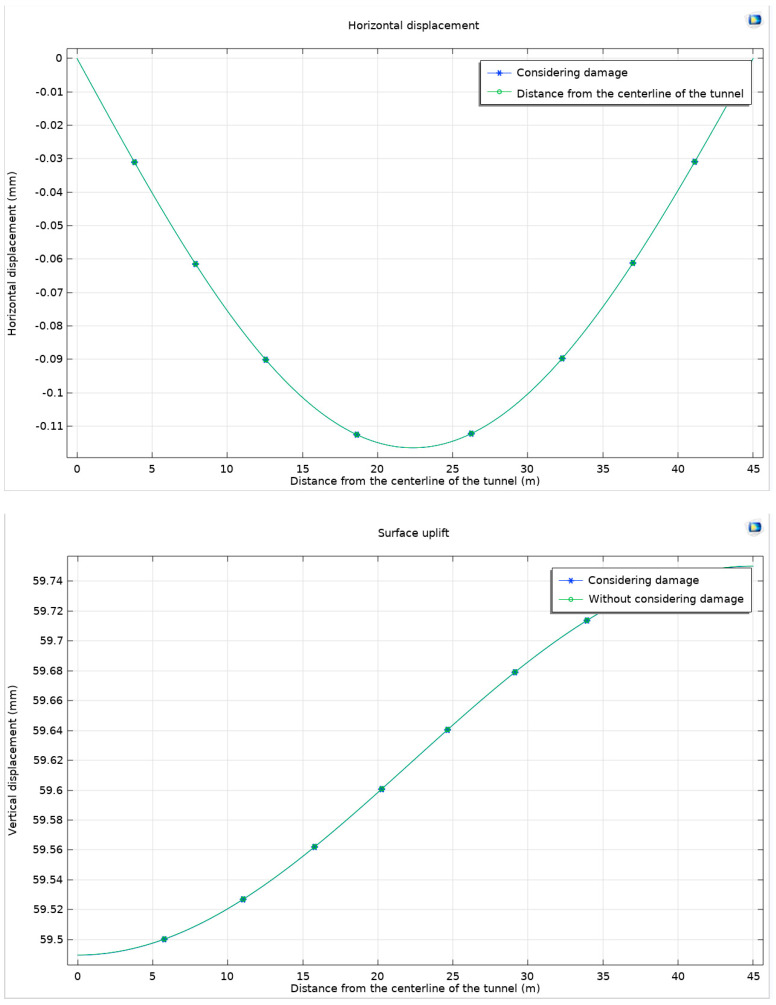
Differences in horizontal displacement and ground uplift and at the maximum value with and without a consideration of damage.

**Figure 27 materials-17-01747-f027:**
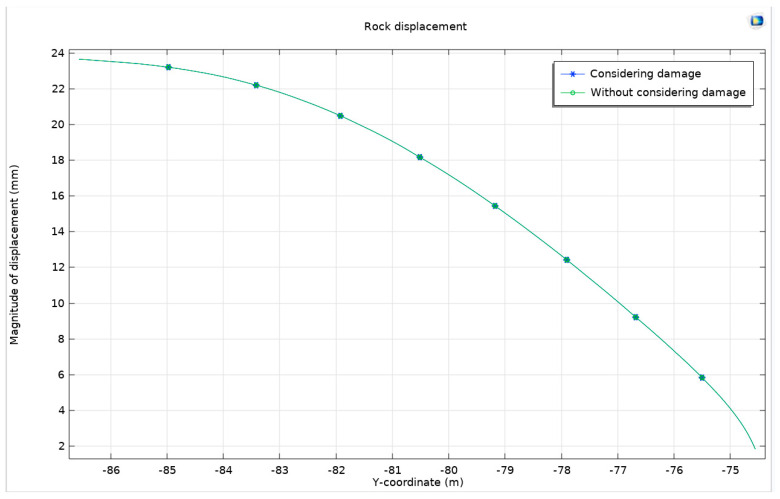
Differences in the displacement of the surrounding rock with and without a consideration of damage.

**Table 1 materials-17-01747-t001:** Model parameters.

*E*/GPa	*c*/MPa	*φ*/°	ρ/ KN · m^−3^	*v*	$k_{w}$/ cm · s^−1^	*α*	$\omega_{max}$	$\omega_{0}$
1.0	0.3	20	17.5	0.35	2 × 10^−10^	0.03	3%	2%

## Data Availability

Data are contained within the article.

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
