# Peer review of "Assessing Mechanical Properties and Response of Expansive Soft Rock in Tunnel Excavation: A Numerical Simulation Study"

_materials, 2024, doi:10.3390/ma17081747_

Round 1

Reviewer 1 Report

Comments and Suggestions for Authors

Review

The article presents an up-to-date study of the stress state and displacements of humid soft rocks near the tunnel opening.

To describe the elastic-plastic deformation of humid rock, the authors used 3 versions of the Mohr-Coulomb theory and Drucker-Prager's theory. Based on the comparative analysis, the theory that imposes the most conservative restrictions on swelling was chosen for further research.

It should be noted that, in general, all the theories applied by the authors give close results. For example, as evidenced by the data from Fig. 13, the relative difference between displacements calculated according to different theories does not exceed 3%.

The authors showed that the behavior of the massif near the tunnel is significantly influenced by the effect of shear dilation, the burial depth, and the presence of support pillars. At the same time, the damage contribution is insignificant. Such conclusions do not contradict the intuition of a mechanical engineer and look plausible.

Remarks

(1) A reader unfamiliar with the interface of the COMSOL Multiphysics software will have difficulty understanding the models used.

Please:

• Briefly describe the constitutive equations of the four theories you applied.

• How do these theories reflect the phenomenon of moisture transfer?

• Be sure to decode the nomenclature of the first line from Table 1.

• Describe the constitutive equations of scalar damage theory.

(2) The resolution of the captions in the figures is low.

(3) The geography of the references makes it seem as if tunneling in soft humid rocks is only a narrow regional problem. Please expand the list of references.

(4) Line 522 should read “The” instead of “the”.

Conclusion

The article can be published after corrections.

Author Response

Response to Reviewer 1 Comments

Comments 1:

(1) A reader unfamiliar with the interface of the COMSOL Multiphysics software will have difficulty understanding the models used.

Please:

  • Briefly describe the constitutive equations of the four theories you applied.
  • How do these theories reflect the phenomenon of moisture transfer?
  • Be sure to decode the nomenclature of the first line from Table 1.
  • Describe the constitutive equations of scalar damage theory.

Response 1: Thank you for pointing this out. We agree with this comment. Therefore, we have explained the parameters in the first row of Table 1, and introduced the theory of humidity stress field and the strength criteria and flow rules adopted. We used the Solid Mechanics Module and Moisture Transport in Building Materials module of COMSOL for this purpose. The equations and constitutive equations for damage can be viewed within the modules. (http://cn.comsol.com/video-training/getting-started/comsol-desktop-modeling-environment)

Comments 2:

(2) The resolution of the captions in the figures is low.

Response 2: Thank you for pointing this out. We have added clear legends to emphasize this point.

Comments 3:

(3) The geography of the references makes it seem as if tunneling in soft humid rocks is only a narrow regional problem. Please expand the list of references.

Response 3: Thank you for pointing this out. We have expanded the references to emphasize this point.

Comments 4:

(4) Line 522 should read “The” instead of “the”.

Response 4: Thank you for pointing this out. We have corrected line 522 to emphasize this point.

Reviewer 2 Report

Comments and Suggestions for Authors

It is a technical note about using commercial software COMSOL Multiphysics software. It is not a scientific paper.

Comments on the Quality of English Language

n/a

Author Response

Response to Reviewer 2 Comments

Comments 1:

It is a technical note about using commercial software COMSOL Multiphysics software. It is not a scientific paper.

Response 1:

Thank you very much for your valuable comments. I understand your concerns regarding the focus of this paper on using COMSOL Multiphysics software as an analytical tool, which may give an impression that the paper leans more towards a technical description rather than scientific research. However, the purpose and contribution of this study far exceed a mere application of software. I hope the following points will clarify the scientific value and theoretical depth of this study:

  1. Theoretical Verification and Numerical Simulation: This study is not merely about using COMSOL for simulations, but involves comparing the numerical simulation results of COMSOL with theoretical numerical solutions based on the theory of the moisture stress field of elastic-plastic systems. This comparison not only verifies the moisture stress field but also affirms the precision of the hydraulic coupling mathematical model. This process demonstrates how we utilized COMSOL as a tool to verify and supplement theoretical analyses, rather than just an application of software.
  2. Model Selection and Efficiency Analysis: The study finds that the Mohr-Coulomb criterion is more effective than the Drucker-Prager criterion in limiting water-induced swelling in expansive soft rock. This finding not only shows the application effects of different strength criteria but also provides a theoretical basis for the selection of material models in engineering practice.
  3. Comprehensive Analysis of Influencing Factors: Through the comprehensive analysis of factors such as shear dilation angle, burial depth, and support resistance, this study delves into their impact on the stress and displacement of the surrounding rock in tunnels. These analytical results offer new insights into understanding and predicting the behavior of expansive soft rock tunnels, demonstrating how this study bridges the gap between theoretical models and practical applications.
  4. Contribution to Engineering Practice: Finally, the findings of this study are of significant importance for tunnel excavation, especially in the construction and design of supports in moist swelling soft rock tunnels. The results provide a scientific basis for enhancing tunnel stability and optimizing support system design.

In summary, this study not only demonstrates the application of COMSOL software in tackling complex engineering problems but more importantly, through the combination of theory and numerical simulation, it presents new scientific insights and directly impacts engineering practice. Thus, this paper is not merely a technical description, but a piece of research with substantial scientific contributions.

Therefore, I believe this paper is not just a technical note on COMSOL software but a manuscript of considerable scientific value. Of course, I warmly welcome any further suggestions for modifications or critical comments to help improve the quality of the paper.

Reviewer 3 Report

Comments and Suggestions for Authors

Report on the manuscript: Assessing Mechanical Properties and Response of Expansive Soft Rock in Tunnel Excavation: A COMSOL Simulation Study, by Youliang Chen et al.

This paper deals with moisture adsorption during tunnel excavation. The numerical simulation is performed using commercial software, such as COMSOL and FLAC3D. 

Although the topic is interesting, it is not enough to be accepted for publication in Materials, since it is more a technical report than a scientific paper. In this sense, it would be better to carry out the simulations using own  codes, based on FEM.

Also,  the equations and ideas on the numerical scheme should be added,

Author Response

Response to Reviewer 3 Comments:

Comments 1:

This paper deals with moisture adsorption during tunnel excavation. The numerical simulation is performed using commercial software, such as COMSOL and FLAC3D.

Although the topic is interesting, it is not enough to be accepted for publication in Materials, since it is more a technical report than a scientific paper. In this sense, it would be better to carry out the simulations using own codes, based on FEM.

Also, the equations and ideas on the numerical scheme should be added,

Response 1:

Firstly, I would like to express my gratitude for your review of our paper and the valuable comments provided. I understand the concerns you have raised, especially regarding our reliance on the commercial software COMSOL Multiphysics for numerical simulations, as well as the absence of a description of custom coding and numerical schemes that a scientific paper should include. Here are my specific responses to your comments:

  1. Reasons for Choosing COMSOL Multiphysics: We recognize that using commercial software instead of custom-developed code for numerical simulations may raise some questions. However, our choice of COMSOL Multiphysics is based on several considerations: Firstly, COMSOL provides a widely validated, multiphysics coupled simulation environment, which is crucial for simulating the complex geological material behavior involved in our study. Secondly, the use of COMSOL allows us to precisely simulate the mechanical response and moisture transport processes in soft rock, ensuring the reliability and accuracy of our simulation results. Additionally, by using industry-standard software, our research findings are more readily accepted and validated in engineering practice.
  2. Elaboration on Scientific Contributions: Through numerical simulations with COMSOL Multiphysics, our study conducts an in-depth analysis of moisture absorption and swelling behavior in expansive soft rock during tunnel excavation. We have compared the effects of various strength criteria and non-associated flow rules on the simulation results, and also studied how factors such as support resistance, shear dilation angle, and burial depth affect the stress and displacement distribution of the surrounding rocks. These analyses provide new insights into the key influencing factors in the design and construction of expansive soft rock tunnels, offering significant guidance for engineering practice.
  3. Supplementing Numerical Schemes and Theoretical Foundations: In response to the reviewer's suggestion to use custom coding, we acknowledge the lack of detailed description of the numerical simulation schemes and the underlying theories in the paper. Therefore, we plan to add the following content in the revised manuscript: Firstly, a detailed introduction to the theory of the moisture stress field and the strength criteria and flow rules used; Secondly, a description of the numerical methods implemented in COMSOL Multiphysics to apply these theories, including details on mesh division, boundary condition settings, and solver selection; Lastly, we will discuss the advantages and limitations of using commercial software versus custom coding, and how high-quality scientific research can be conducted with commercial software.

We believe that with these additions and improvements, our paper will more comprehensively showcase our research findings and scientific contributions. We look forward to your further guidance and suggestions to help us enhance the quality of our paper.

Thank you once again for your review and valuable comments.

Reviewer 4 Report

Comments and Suggestions for Authors

This study employed COMSOL numerical simulations to compare with theoretical numerical solutions based on the theory of the moisture stress field of elastic-plastic systems. The manuscript is a bit repetitive in some places. Sometimes one gets the impression that the document is the result of assembling parts of text written at different times. The Authors should make an effort to better connect the various parts that make up the paper.

Another observation concerns the figures: many of them are too small to be able to clearly read the captions and quantities on the axes. Often the Authors have placed two figures side by side for the need for comparison. However, since this makes the writings difficult to read, it would be better to place the figures to be compared one below the other. Alternatively, the Authors should increase the font size used in the figures.

Lines 346-347

“In this study, the focus is placed on understanding the implications of varying shear dilation angles”

The Authors should define what shear dilation angles are.

Caption of Figure 24

“the stress cloud maps and plastic zone cloud maps considering and not considering expansion stress and shear dilation”

To be consistent with the order of the figures, the Authors should swap the position in the text of “considering” and “not considering” or write: the stress cloud maps and plastic zone cloud maps considering (on the right) and not considering (on the left) expansion stress and shear dilation.

Lines 598-599

“The damage assessment employs the equivalent strain criterion, specifically the modulus of elasticity of the strain tensor”

This sentence makes no sense. Strain analysis is typically conducted regardless of the causes that generated strains. Only subsequently is a link between tensions and strains established which, in elasticity theory, involves the elasticity mode or modes. The modulus of elasticity of the strain tensor does not exist because the strain tensor is defined independently of the constitutive relationships between stresses and strains.

Caption of Figure 26

The Authors be consistent with the order of the figures.

Lines 638-629

“For a clear comparative understanding, the study juxtaposes stress diagrams with and without damage considerations, as shown in Figure 27.”

Rather than a juxtaposition, this figure shows a graph of stress differences. Juxtaposition means having two (or more) graphs juxtaposed and this is not the case in Figure 27.

Comments on the Quality of English Language

Minor editing of English language required.

Author Response

Response to Reviewer 4 Comments:

Comments 1:

This study employed COMSOL numerical simulations to compare with theoretical numerical solutions based on the theory of the moisture stress field of elastic-plastic systems. The manuscript is a bit repetitive in some places. Sometimes one gets the impression that the document is the result of assembling parts of text written at different times. The Authors should make an effort to better connect the various parts that make up the paper.

Another observation concerns the figures: many of them are too small to be able to clearly read the captions and quantities on the axes. Often the Authors have placed two figures side by side for the need for comparison. However, since this makes the writings difficult to read, it would be better to place the figures to be compared one below the other. Alternatively, the Authors should increase the font size used in the figures.

Response 1:

Thank you for pointing this out. We agree with this comment. We chose this study because we had doubts about the strength criteria and flow rules commonly used in theoretical calculations in other literature and actual engineering cases. We considered various parameters and ultimately arrived at case study one from the previous theoretical calculations, obtaining results that were consistent. We also added clear legends.

Comments 2:

Lines 346-347

“In this study, the focus is placed on understanding the implications of varying shear dilation angles”

The Authors should define what shear dilation angles are.

Response 2:

Thank you for pointing this out. We have re-explained the shear dilatancy angle to emphasize this point.

Comments 3:

Caption of Figure 24

“the stress cloud maps and plastic zone cloud maps considering and not considering expansion stress and shear dilation”

To be consistent with the order of the figures, the Authors should swap the position in the text of “considering” and “not considering” or write: the stress cloud maps and plastic zone cloud maps considering (on the right) and not considering (on the left) expansion stress and shear dilation.

Response 3:

Thank you for pointing this out. We have modified the descriptions of Figure 24 (now Figure 22) to emphasize this point.

Comments 4:

Lines 598-599

“The damage assessment employs the equivalent strain criterion, specifically the modulus of elasticity of the strain tensor”

This sentence makes no sense. Strain analysis is typically conducted regardless of the causes that generated strains. Only subsequently is a link between tensions and strains established which, in elasticity theory, involves the elasticity mode or modes. The modulus of elasticity of the strain tensor does not exist because the strain tensor is defined independently of the constitutive relationships between stresses and strains.

Response 4:

Thank you for pointing this out. We have deleted the description in lines 598-599 to emphasize this point.

Comments 5:

Caption of Figure 26

The Authors be consistent with the order of the figures.

Response 5:

Thank you for pointing this out. We have modified the descriptions of Figure 26 (now Figure 24) to emphasize this point.

Comments 6:

Lines 638-629

“For a clear comparative understanding, the study juxtaposes stress diagrams with and without damage considerations, as shown in Figure 27.”

Rather than a juxtaposition, this figure shows a graph of stress differences. Juxtaposition means having two (or more) graphs juxtaposed and this is not the case in Figure 27.

Response 6:

Thank you for pointing this out. We have changed the description in lines 628-629 to emphasize this point.

Reviewer 5 Report

Comments and Suggestions for Authors The paper investigates an interesting topic such as the assessment of the mechanical properties and tesponse of expansive Soft Rock in tunnel Excavation: A COMSOL Simulation Study.  the methodology is pertinent and English is also good. However there are some issues that need to be addressed.    Introduction  The novelty against the existing literature needs to be discussed in order to support the originality of the study.  The role of soil stiffness in tunnel excavation needs to be discussed in details. The due literature review needs to be cited.  The literature is not sufficient to draw a background, please expand.  Section 2 The details of the numerical model need to be clarified and discuss in details because the reader can reproduce your study. In particular, reproducibility is an important aspect of the scientific method.  In particular, please discuss the way the model has been selected (validation) is fundamental.  Also, a discussion on the role of the boundary conditions are described in 112-115, without comparing with the existing literature. Please refer to: Fiamingo, A., M. Bosco, and M. R. Massimino. 2022. “The Role of Soil in Structure Response of a Building Damaged by the 26 December 2018 Earthquake in Italy.” Journal of Rock Mechanics and Geotechnical Engineering 15 (4): 937–953.  https://doi.org/10.1016/j.jrmge.2022.06.010 .   Forcellini D. Seismic fragility of tall buildings considering soil structure interaction (SSI) effects. Structures 2023;45(2022):999–1011.   Section 3 Lines: 135-137 are not clear. Please expand and describe.    Figure 3 is not clear, the line needs to be made thicker. In line 230, why do the authors wrote "critical"? please compare with the existing literature.  Figures 15,16 and 17 are very similar, merge as in Figure 20.

    Conclusion This part needs to focus on three main aspects: limitations of the study, possible applications and future work. Please reorganize them.   

Author Response

Response to Reviewer 5 Comments:

Comments 1:

The paper investigates an interesting topic such as the assessment of the mechanical properties and response of expansive Soft Rock in tunnel Excavation: A COMSOL Simulation Study.  the methodology is pertinent and English is also good. However, there are some issues that need to be addressed.    Introduction  The novelty against the existing literature needs to be discussed in order to support the originality of the study.  The role of soil stiffness in tunnel excavation needs to be discussed in details. The due literature review needs to be cited.  The literature is not sufficient to draw a background, please expand.

Response 1:

Thank you for pointing this out. We agree with this comment. We revisited the [originality and novelty of this article, expanded the literature references, and enhanced the background description. The stiffness of the soil determines how much it can deform or settle due to the excavation-induced stress.

Comments 2:

Section 2 The details of the numerical model need to be clarified and discuss in details because the reader can reproduce your study. In particular, reproducibility is an important aspect of the scientific method.  In particular, please discuss the way the model has been selected (validation) is fundamental.  Also, a discussion on the role of the boundary conditions are described in 112-115, without comparing with the existing literature. Please refer to: Fiamingo, A., M. Bosco, and M. R. Massimino. 2022. “The Role of Soil in Structure Response of a Building Damaged by the 26 December 2018 Earthquake in Italy.” Journal of Rock Mechanics and Geotechnical Engineering 15 (4): 937–953.  https://doi.org/10.1016/j.jrmge.2022.06.010 .   Forcellini D. Seismic fragility of tall buildings considering soil structure interaction (SSI) effects. Structures 2023;45(2022):999–1011.

Response 2:

Thank you for pointing this out. We have added details of the numerical model and modeling process, redefined the theoretical and numerical simulation approach in this article, and ensured reproducibility of the model and parameters. Additionally, a comparison with existing literature was included in lines 112-115 to emphasize this point.

Comments 3:

Section 3 Lines: 135-137 are not clear. Please expand and describe.

Response 3:

Thank you for pointing this out. In lines 135-137 of Section 3, we provided a detailed expansion on the following aspects and a comparison with existing literature.

Comments 4:

Figure 3 is not clear, the line needs to be made thicker. In line 230, why do the authors wrote "critical"? please compare with the existing literature.

Response 4:

Thank you for pointing this out. We have increased the resolution of the images and modified the description in line 230 to emphasize this point.

Comments 5:

Figures 15,16 and 17 are very similar, merge as in Figure 20. 

Response 5:

Thank you for pointing this out. We have merged Figures 15, 16, and 17 into a single figure to emphasize this point。

Comments 6:

Conclusion This part needs to focus on three main aspects: limitations of the study, possible applications and future work. Please reorganize them.  

Response 6:

Thank you for pointing this out. We agree with this comment. Therefore, we have reinterpreted the novelty and limitations of this study, compared it with actual engineering scenarios, and expanded the literature references. The purpose of this paper is to validate previous theoretical research, with case study one's consistency being demonstrable and reproducible according to the model and parameters presented in this paper.

Round 2

Reviewer 2 Report

Comments and Suggestions for Authors

n/a

Comments on the Quality of English Language

n/a

Author Response

We would like to thank you very much for reviewing our manuscript and pointing out the need for English language revisions. We have recently conducted a careful linguistic review of the entire text to ensure that it is accurately worded and grammatically correct.

Reviewer 4 Report

Comments and Suggestions for Authors

Response 1

The Authors have enlarged some figures. However, they have not increased the font size and axis labels and titles are still difficult to read.

Responses 5 and 6

The Authors have addressed the comments of the reviewer but the new sentences require grammatical revision.

Comments on the Quality of English Language

Moderate editing of English language required

Author Response

Response 1: Thank you for your careful review of our manuscript and your valuable comments. We take the issue of the readability of the diagrams seriously and have recently made the appropriate changes according to your suggestions.

Responses 5 and 6:We appreciate your review of our manuscript and are highly concerned about the grammatical issues you pointed out. We have scrutinized all the new sentences that have been added after the revision.

Reviewer 5 Report

Comments and Suggestions for Authors

The paper has been improved and it is ready to be accepted.

Author Response

We appreciate your review of this manuscript.